# Cost-effectiveness of Spironolactone for Adult Female Acne (SAFA): economic evaluation alongside a randomised controlled trial

Sarah Pyne ,[1] Tracey H Sach ,[1,2] Megan Lawrence,[3] Susanne Renz,[3] Zina Eminton,[3] Beth Stuart ,[2,4] Kim S Thomas ,[5] Nick Francis ,[2] Irene Soulsby,[6] Karen Thomas,[6] Natalia V Permyakova,[3] Matthew J Ridd ,[7] Paul Little ,[2] Ingrid Muller ,[2] Jacqui Nuttall,[3] Gareth Griffiths,[3] Alison M Layton,[8] Miriam Santer [2]

For numbered affiliations see end of article.

**Correspondence to**
Dr Tracey H Sach;
t.sach@soton.ac.uk

## ABSTRACT

**Objective** This study aims to estimate the cost-effectiveness of oral spironolactone plus routine topical treatment compared with routine topical treatment alone for persistent acne in adult women from a British NHS perspective over 24 weeks.

**Design** Economic evaluation undertaken alongside a pragmatic, parallel, double-blind, randomised trial.

**Setting** Primary and secondary healthcare, community and social media advertising.

**Participants** Women ≥18 years with persistent facial acne judged to warrant oral antibiotic treatment.

**Interventions** Participants were randomised 1:1 to 50 mg/day spironolactone (increasing to 100 mg/day after 6 weeks) or matched placebo until week 24. Participants in both groups could continue topical treatment.

**Main outcome measures** Cost-utility analysis assessed incremental cost per quality-adjusted life year (QALY) using the EQ-5D-5L. Cost-effectiveness analysis estimated incremental cost per unit change on the Acne-QoL symptom subscale. Adjusted analysis included randomisation stratification variables (centre, baseline severity (investigator's global assessment, IGA <3 vs ≥3)) and baseline variables (Acne-QoL symptom subscale score, resource use costs, EQ-5D score and use of topical treatments).

**Results** Spironolactone did not appear cost-effective in the complete case analysis (n=126 spironolactone, n=109 control), compared with no active systemic treatment (adjusted incremental cost per QALY £67 191; unadjusted £34 770). Incremental cost per QALY was £27 879 (adjusted), just below the upper National Institute for Health and Care Excellence's threshold value of £30 000, where multiple imputation took account of missing data. Incremental cost per QALY for other sensitivity analyses varied around the base-case, highlighting the degree of uncertainty. The adjusted incremental cost per point change on the Acne-QoL symptom subscale for spironolactone compared with no active systemic treatment was £38.21 (complete case analysis).

## STRENGTHS AND LIMITATIONS OF THIS STUDY

⇒ Our study is based on individual patient-level data collected alongside the first large pragmatic, parallel, double-blind, randomised trial of spironolactone for acne.

⇒ In addition to the base-case analysis seeking to answer the question of whether spironolactone is cost-effective compared with no active systemic treatment (both groups could use routine topical treatments) in women with persistent acne, a number of sensitivity analyses were undertaken to provide a range on estimates of cost-effectiveness under different scenarios.

⇒ Differential rates of missing data between groups over time were addressed by undertaking both a complete case analysis and multiple imputation to explore the impact of missing data on the study conclusions.

⇒ As the study was constrained by the design of the clinical trial, the base-case did not reflect real-world prescribing in the comparator group, limiting interpretation of the results.

⇒ The results reflect the method of data collection and may have been limited as a consequence of resource-use under-reporting, short time-frame and limited sensitivity of the EQ-5D outcome measure in patients with acne.

**Conclusions** The results demonstrate a high level of uncertainty, particularly with respect to estimates of incremental QALYs. Compared with no active systemic treatment, spironolactone was estimated to be marginally cost-effective where multiple imputation was performed but was not cost-effective in complete case analysis.

**Trial registration number** ISRCTN registry (ISRCTN12892056).

## INTRODUCTION

Acne (acne vulgaris) is a common condition, affecting >80% of people at some point

in their life.[1] Its impact on the NHS is considerable, being responsible for around 3.5 million consultations with a General Practitioner (GP)[1] and 70 000 referrals for specialist care[2] in the UK annually. As well as direct burdens to the NHS, adults (18–30 years) with severe acne in the UK have higher unemployment rates[3] and a small study by Jowett and Ryan[4] showed that 45% (13/29) of acne patients reported interpersonal difficulties at work.

There are many treatment options for women with moderate-to-severe acne, but a recent network meta-analysis (NMA) demonstrated paucity of good-quality evidence and the complexity of choice.[5] Informed in large part by this NMA and the associated economic model,[6] the National Institute for Health and Care Excellence (NICE) guidelines on the management of acne vulgaris recommend a fixed combination topical preparation containing retinoids, benzoyl peroxide or antibiotics as first-line treatment for any severity of acne, while a fixed combination topical agent plus oral lyme-cycline or doxycycline once daily is recommended for moderate-to-severe acne. The latter is also recommended for moderate-to-severe acne that does not respond adequately to a 12-week course of treatment that does not include an oral antibiotic.[7] The guidance states that treatment options including an antibiotic (topical or oral) should only be continued for more than 6 months in exceptional circumstances (other guidelines limit oral antibiotic duration to 3 months)[8–10] and that clinicians should be aware of the associated risks of antimicrobial resistance. Doctors, however, report many challenges when trying to discontinue oral antibiotics.[11]

Spironolactone is already used off license for women with acne, is an inexpensive treatment choice and could play a role in reducing antibiotic use.[12] Literature searches did not, however, find any previously published economic evaluations on the cost-effectiveness of spironolactone in this group of patients, although there are two other ongoing studies of spironolactone in France and the USA, the former of which includes an economic evaluation.[13 14] In this paper, we estimate the cost-effectiveness of spironolactone plus routine topical treatment compared with no active systemic treatment plus routine topical treatment for persistent acne in adult women from a British NHS perspective over 24 weeks.

## PATIENTS AND METHODS

The Spironolactone for Adult Female Acne (SAFA) trial was a pragmatic, multicentre, participant-led and clinician-blind, superiority, randomised trial with two parallel treatment groups: spironolactone compared with placebo in women aged 18 years and older with facial acne judged to warrant oral antibiotics. The economic evaluation was nested within this trial.

Participants were recruited in primary care, secondary care and through advertising (community and social media). Baseline assessment was conducted by a research nurse and/or dermatologist in secondary care clinics to ensure standard clinical assessments, as the investigator's global assessment (IGA) for acne was an inclusion criterion and an important secondary outcome. Baseline appointments included a pregnancy test, blood test (to exclude renal impairment or raised serum potassium), participant photo to aid recall about changes in acne and contraceptive counselling. The first participant was recruited in June 2019 and the last in August 2021, while follow-up finished February 2022. The SAFA trial is described in more detail in the clinical paper.[15 16]

Participants were randomised 1:1 using online software to either 50 mg/day spironolactone or matched placebo until week 6, increasing to 100 mg/day spironolactone or matched placebo until week-24, assuming treatment was tolerated. Participants were stratified by recruitment centre and baseline acne severity (IGA<3 vs IGA≥3). In both groups participants could continue using topical treatment. Between baseline and week 12, participants were asked not to take oral treatment for acne other than study medication, except for oral contraception taken for over 3 months previously. After 12 weeks, participants in both groups could receive usual care, including oral treatments, such as oral antibiotics, hormonal treatment or isotretinoin. In both groups, participants were followed up face-to-face (or by video call or telephone due to COVID-19) at week 6 and week 12 in secondary care, with primary outcome assessment at week 12, and longer term follow-up by questionnaires at week-24.

Although in the clinical trial, spironolactone plus routine topical treatment was compared with placebo plus routine topical treatment, it is most appropriate in economic evaluations to compare an active treatment to current usual care.[17] Therefore, to use the data collected in the trial while reflecting a useful analysis to decision makers in practice, this economic evaluation compared spironolactone plus routine topical treatment to no active systemic treatment plus routine topical treatment.

### Measuring costs

In keeping with an NHS perspective, all acne-related resource use data, including intervention, primary and secondary care visits, and prescription medication use, were collected for participants in both groups. Personal Social Services resource use was not collected, as patient and clinician contributors did not anticipate these being incurred by participants.

Resource use data were collected via case report forms and participant questionnaires (see online supplemental file 2 for a copy), designed with the input of public contributors, at baseline (collecting the preceding 6 weeks), week 6, week 12 and week-24 for the intervention phase.

Resource use was valued using UK unit costs (£ Sterling) for the most current price year available at the start of analysis (financial year 2021) and identified from published sources.

The intervention was costed as described in figure 1, which assumes that standard treatment

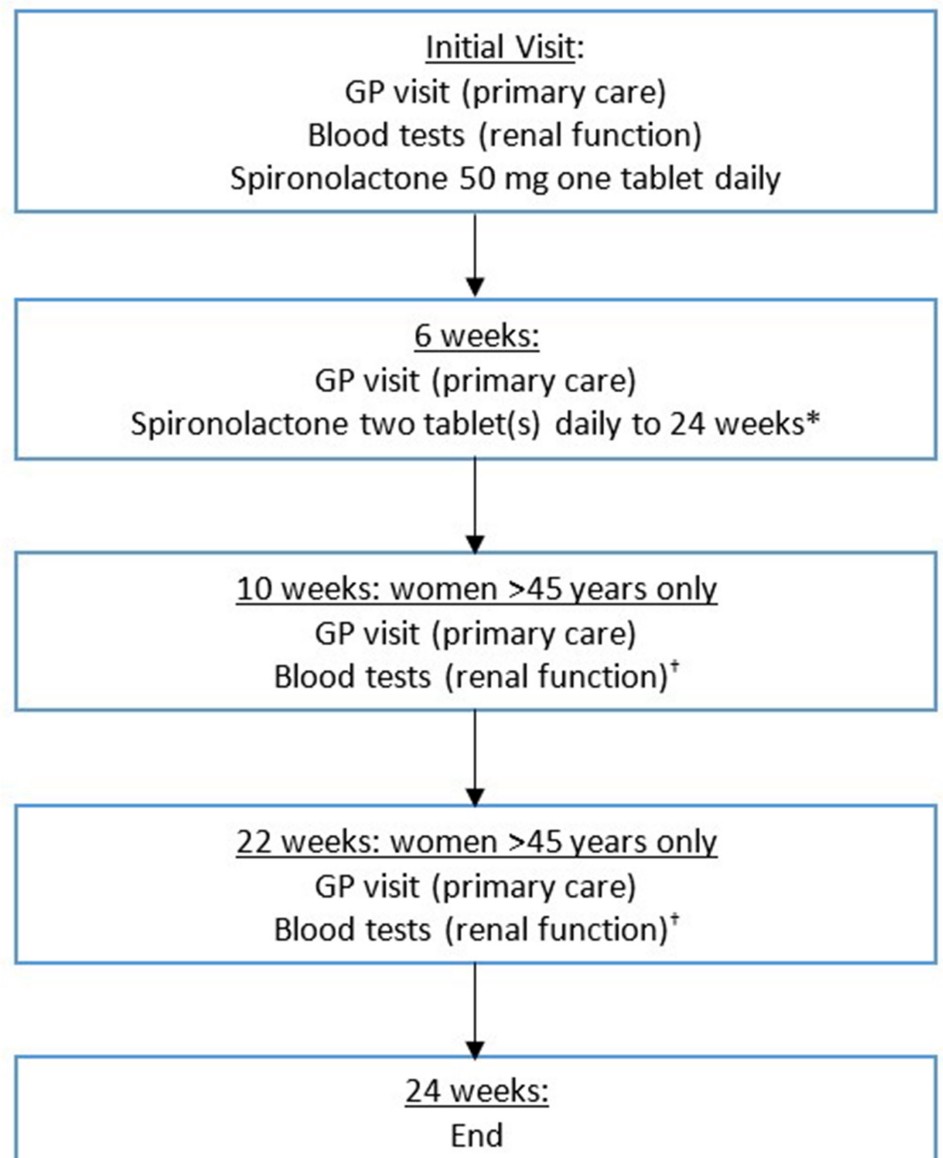

**Figure 1** Intervention resource use as per standard treatment with spironolactone (base-case).* Assumes all patients escalated to two 50 mg tablets spironolactone at 6-week visit. Based on the data from the trail, this was the case for 182/184 (99%) in the spironolactone group at 6 weeks (question response rate 184/202 in spironolactone group).† Existing evidence and expert opinion recommend ongoing blood monitoring for women aged >45 years, or those with relevant comorbidities or on treatments with increased risk. As the latter two were not included in the trial, it is not possible to estimate the proportion of such patients that might receive spironolactone and need blood test monitoring. 6/201 (30%) patients in the spironolactone arm of the trial were aged >45 years. GP, General Practitioner.

with spironolactone, if adopted, will be delivered in primary care, including two GP visits (unless >45 years of age), baseline blood test and the cost of spironolactone (50 mg 6 weeks, 100 mg 18 weeks).[10 18–20] No intervention costs (placebo tablets, GP visits to prescribe placebo tablets or blood tests) were included for the no active systemic treatment group as these would not occur if no intervention was being given (the comparator for this economic evaluation).

Acne-related resource use data related to visits to community-based healthcare professionals (HCP), visits to hospital out-patient and in-patient services (including accident and emergency) and prescribed medication costs were self-reported via participant questionnaires at all time points, including baseline for participants in both groups. When asked about medication use, participants were asked to report only what they had been prescribed since the previous follow-up visit. Unit costs for each visit-type were combined with this data to estimate the total community-based HCP visit costs and the total hospital contact costs. Participants were also asked for details of prescribed acne-related medication including type, strength and quantity. Unit costs for all medication types[21] were used to estimate the prescription costs over the 24-week treatment period.

The mean (SD) cost per participant per intervention group was estimated for the 24-week treatment period, for each of the cost types described above and mean difference (95% CI) in NHS cost was estimated.

### Measuring outcomes

The primary economic outcome measure was quality-adjusted life years (QALYs) over the trial period of 24 weeks, as measured by the generic preference-based EQ-5D-5L questionnaire.[22] Responses were converted to utility scores using the EQ-5D-5L Crosswalk UK preference weights, as this was in line with recommendations at the point analysis started, where utility ranges from −0.594 to 1.[23 24] Utility values were used to estimate QALYs over 24 weeks, using both linear interpolation and area under the curve analysis.[25]

A secondary economic outcome was the Acne-QoL symptom subscale score (five questions with seven responses to each)[26 27] at week 24, used as an estimate of effectiveness, which enables comparison with future economic studies in acne.

### Economic analysis

The base-case cost-utility analysis (CUA) and secondary cost-effectiveness analysis (CEA) incorporated all randomised participants with complete cost and outcome data. Given the 24-week time-horizon, costs and benefits were not discounted.[24]

The base-case CUA estimated the incremental cost per QALY (incremental cost-effectiveness ratio, ICER) to enable comparison with the cost-utility of other interventions. The incremental cost (95% CI) and QALY change (95% CI) between groups was estimated unadjusted and adjusted for randomisation stratification variables (centre, baseline severity (IGA <3 vs ≥3)) and baseline variables (including Acne-QoL symptom subscale score, resource use costs, EQ-5D score and use of topical treatments (Y/N)). In line with NICE guidance,[24] we estimated whether the intervention was cost-effective by comparing the ICER with a cost-effectiveness threshold of £20 000 to £30 000 per QALY.

A CEA estimated the incremental cost per unit change on the Acne-QoL symptom subscale score. The incremental cost (95% CI) and Acne-QoL symptom subscale change (95% CI) between groups was estimated unadjusted and adjusted as described for the base-case CUA. The CUA and CEA were undertaken using a regression-based approach (seemingly unrelated regression equations).[28] Published guidelines for the economic evaluation of healthcare interventions were followed as appropriate.[29 30]

To estimate the level of uncertainty associated with the decision regarding cost-effectiveness, Fieller's theorem was used to calculate[31] the probability of being cost-effective at the £20 000 and £30 000 willingness-to-pay threshold values.[24] Non-parametric bootstrapping was conducted to generate 10 000 estimates of incremental costs and benefits. From this, cost-effectiveness acceptability curves (CEACs) were generated to show the probability that the intervention is cost-effective at different willingness-to-pay values.

Several sensitivity analyses were agreed and specified in the health economic analysis plan (HEAP) before analysis to explore key uncertainties around important parameters in the economic evaluation. The impact of missing data on cost-effectiveness estimates was explored by undertaking multiple imputation (MI) (SA1), assuming that the data were missing at random and using chained equations to handle the missing cost and outcome data.[31] Second, the impact of costing the intervention as per the SAFA trial protocol (ie, intervention was accessed via secondary care, excluding any research related costs) was explored (SA2). The cost utility analysis was repeated but with the intervention costed as described in online supplemental figure S1, while the placebo group was costed as in the base-case analysis, that is, assumed no intervention costs. Third, the CUA was repeated assuming that, as this patient population had persistent acne of sufficient severity to warrant treatment with oral antibiotics, all women in the no active systemic treatment group took oral antibiotics (lymecycline or doxycycline, 1 tablet daily for 24 weeks) as per NICE guidance,[32] in addition to topical treatment (SA3). To cost this intervention, the weighted mean cost per dose of doxycycline/lymecycline was used (see table 1) and two GP visits were assumed. Due to a lack of evidence about the incremental QALYs between spironolactone plus topical treatment versus oral antibiotics plus topical treatment, a threshold analysis was performed to ascertain what level of incremental QALYs would switch the intervention between cost-effective and not cost-effective. Incremental costs (95% CI) and the threshold value for incremental QALYs are presented in the results. Potential costs associated with antibiotic-related side effects and the societal costs of over prescribing of oral antibiotics were not included. Finally, a sensitivity analysis exploring a wider perspective than that limited to the NHS was conducted (SA4). In addition to NHS-related resource use data, the following was collected via participant questionnaire: out-of-pocket expenses (including, complementary therapist visits, cosmetic skin care products, non-NHS-prescribed medication, parking and travel costs for healthcare appointments and other) and productivity losses (including lost patient and carer productivity). These were valued using participant self-reported values and unit costs identified from published sources, as reported in table 1, and summed along with NHS costs to estimate the mean difference (95% CI) in total costs (wider perspective). Utility analysis was then repeated as described for the base-case. A subgroup analysis based on age was also conducted and is presented in online supplemental appendix S2.

**Table 1** Unit costs (UK£ sterling, 2020/2021 financial year)

| Cost item | Unit cost (£) | Unit | Source, assumptions |
|---|---|---|---|
| **Intervention** | | | |
| Spironolactone with dose escalation | £49.37 | Total | Prescription Cost Analysis 2021[21] |
| GP visit related to intervention | £33.00 | Total | PSSRU Unit costs 2021[37] |
| Blood test for renal function (eGFR) and potassium level (K serum) | £5.22 | Total | National Cost Collection 2020[38]* |
| **Medication costs** | Mean cost per quantity | | |
| Topical preparations for acne | £0.96 | gram/mL | Prescription Cost Analysis 2021[21] |
| Other topical preparation | £0.03 | gram/mL | Mean across all medications in each medication type. Weighted averages taken where listed >1 x. |
| Oral contraceptives | £0.08 | Tablet | Weighted average for estimating oral antibiotic control for SA (see table 3). Assumes 1×100 mg (doxycycline)/408 mg (lymecycline) per day for 24 weeks. |
| Oral antibiotics | £0.22 | Capsule/tablet | |
| Anti-depressants | £0.20 | Capsule/tablet | |
| Analgesics | £0.04 | Capsule/tablet | |
| PCOS/diabetes medication | £0.03 | Tablet | |
| Other medications | £0.40 | Various | |
| Doxycycline/lymecycline weighted average | £0.25 | Capsule | |
| **Community-based HCP contacts** | | | |
| GP visit unrelated to intervention | £33.00 | Visit | PSSRU Unit costs 2021.[37] |
| Practice Nurse | £14.13 | Visit | PSSRU Unit costs 2021 and 2015.[37 39] |
| NHS Walk-in centre | £71.99 | Visit | National Cost Collection 2020.[38] Weighted average of all community health services.* |
| Community dermatology service | £121.01 | Visit | National Cost Collection 2020.[38]* |
| Healthcare assistant | £14.44 | Visit | PSSRU Unit Costs 2021[37] and UKHCA Commissioning Survey 2012.[40] |
| Pharmacist | £6.99 | Visit | PSSRU Unit costs 2021 and 2015[37 39] and PSNC Pharmacy Advice Audit 2021.[41] |
| Physiotherapist | £66.82 | Visit | National Cost Collection 2020.[38]* |
| Dietician | £82.46 | Visit | National Cost Collection 2020.[38]* |
| Other (community) | £33.00 | Visit | PSSRU Unit costs 2021. Used most common visit: GP visit.[37] |
| **Hospital out-patient contacts** | | | |
| Dermatologist | £128.25 | Visit | National Cost Collection 2020.[38]* |
| Dermatology nurse | £100.71 | Visit | National Cost Collection 2020.[38]* |
| Ear, nose and throat (ENT) | £116.11 | Visit | National Cost Collection 2020.[38]* |
| Interventional radiology | £137.64 | Visit | National Cost Collection 2020.[38]* |
| Trauma and orthopaedics | £125.67 | Visit | National Cost Collection 2020.[38]* |
| Respiratory medicine | £161.07 | Visit | National Cost Collection 2020.[38]* |
| Other (out-patient) | £137.10 | Visit | National Cost Collection 2020.[38]* |
| **Hospital admission** | | | |
| Accident and emergency | £182.28 | Visit | National Cost Collection 2020. Index/Accident & Emergency.[38]* |
| **Wider costs** | | | |
| Personal out-of-pocket expenses | Various | Per item | Participant reported. |
| Lost work time | £18.01 | Hour | ONS 2021[42] Mean hourly earnings, excluding overtime (£). |

*Inflated to 2021 prices as per NHSCII Pay & Prices.[37]
GP, General Practitioner; HCP, Healthcare Professional; ONS, Office for National Statistics; PCOS, Polycystic Ovary Syndrome; PSSRU, Personal Social Services Research Unit.

Stata MP V.17 was used to conduct the analyses. A HEAP was written and followed; a copy is available from the corresponding author.

**Patient and public involvement**
Key questions relating to research design were explored with a virtual acne-specific patient panel and patient survey carried out via the UK Dermatology Clinical Trials Network. Two public contributors (IS and KaT) with experience of acne were members of the trial management group as part of this role they helped identify relevant resources and outcomes and how this data should be collected. They

also contributed to the interpretation and write-up of the health economics component.

## RESULTS
### Participant characteristics
The clinical trial results, including details on sample size and participant characteristics, are reported elsewhere.[16] Of the 410 women recruited to the trial, 201 were randomly assigned to spironolactone and 209 allocated to placebo at the start of the trial. All were allowed to continue routine topical treatment. At week 24, 126 women in the spironolactone group and 109 women in the placebo group had complete cost and outcome data, and these formed the base-case unadjusted CUA. Mean age was 29.2 years, mean BMI was 26.1, at baseline 83% (340/410) participants were using or had used topical treatments, and the majority (75% (306/410)) had acne for two or more years. There were no significant differences in characteristics between groups.[16]

### Costs
The unit costs used in the analysis are presented in table 1. The levels of resource use in each group were very similar prior to randomisation (online supplemental table S1).

The majority of responding women in the spironolactone group (182/184, 99%) increased to two tablets of spironolactone at week 6. The 'standard treatment' approach, used in the base-case economic evaluation, gave rise to a mean total intervention resource use cost of £122.87 (SD £13.04) per participant in the spironolactone group (table 2).

Using available case data, when intervention use was combined with other health resource use, the unadjusted mean incremental cost per participant was £126.35 (95% CI £112.88 to £139.82) for women receiving spironolactone compared with women receiving no active systemic treatment in the base-case (table 2). Excluding intervention costs, the difference was not significant between groups. While patients were asked about in-patient visits, none was reported.

### Outcomes
The mean (SD) QALYs over 24 weeks in the spironolactone group were 0.417 (0.058) per participant compared with 0.404 (0.079) per participant in the no active systemic treatment group, giving an incremental difference of 0.013 (95% CI −0.0024 to 0.0289) QALYs using unadjusted available case data (table 2). The wide 95% CIs around mean estimates demonstrate a high degree of uncertainty.

The mean (SD) change from baseline in Acne-QoL symptom subscale score at 24 weeks was 8.15 (6.12) in the spironolactone group compared with 4.46 (6.34) in the no active systemic treatment group. Thus, the incremental difference in score was 3.68 (95% CI 2.26 to 5.11) in favour of the spironolactone group (table 2).

### Base-case cost utility analysis
In the complete case analysis (CCA), the incremental cost for the spironolactone group (n=118) compared with the no active systemic treatment group (n=101) was £125.36 (95% CI £111.13 to £139.58) (unadjusted this was £125.53 (95% CI £112.15 to £138.91)) (table 3). The adjusted incremental QALYs for the spironolactone group compared with the no active systemic treatment group were 0.0019 (95% CI −0.0096 to 0.0133) (unadjusted was 0.0036, 95% CI −0.0117 to 0.0189). The ICER was £67 191 (unadjusted £34 770) per QALY. At a willingness to pay of £30 000 per QALY, there was a 35% (unadjusted 47%) chance of spironolactone being cost-effective in this population of women with persistent acne.

The CEACs (figure 2), of the adjusted and unadjusted base-case analysis, show that the probability of spironolactone being cost-effective only approaches 50% as the threshold value approaches £120 000 (adjusted), demonstrating a high degree of uncertainty associated with the decision under these conditions.

### Secondary cost-effectiveness analysis
The adjusted incremental difference in cost per point change on the Acne-QoL symptom subscale for the spironolactone group (n=119) compared with no active systemic treatment group (n=102) was £38.21 (unadjusted £35.91) based on a CCA (table 3). How much a decision-maker would be willing to pay for a point change on the Acne-Qol symptom subscale is unknown.

### Sensitivity analyses
The results of the sensitivity analyses are presented in table 3 and prove influential to the conclusions reached. The ICER varies around the base-case from £27 879 (with a 53% probability of being cost-effective at £30 000 threshold) for the MI analysis (SA1) to spironolactone being dominated (more costly and less effective than control) for the wider perspective (CCA) analysis.

There were differential rates of attrition with greater missing data in the no active systemic treatment group, compared with spironolactone group, by 24 weeks follow-up, for costs (39% vs 24%, respectively) and EQ-5D-5L (33% vs 20%, respectively). This may offer some explanation for why, when using MI in a sensitivity analysis, the ICER was less than in the complete case, adjusted analysis (table 3).

With regards to the oral antibiotic control analysis (SA3), the planned threshold analysis using the complete case, adjusted data found that the incremental QALY benefit for spironolactone compared with oral antibiotics would have to be 0.00057 (0.000384, MI adjusted) or less, over 24 weeks, for spironolactone to be less cost-effective than oral antibiotics at a £30 000 threshold. The plausibility of this

**Table 2** Estimates of mean change in resource use and cost (UK£ 2021/22) and mean utility and QALY gain by treatment group (based on available case data)

| Resource | Spironolactone (N=201) | | No active systemic treatment (N=209) | | Mean difference |
| --- | --- | --- | --- | --- | --- |
| | Mean (n) | SD | Mean (n) | SD | (95% CI) |
| **Resource use over 24-week period:** | | | | | |
| Spironolactone (number) | 294 (201) | 0 | 0 (209) | 0 | – |
| GP visits related to intervention (number of visits)* | 2.06 (201) | 0.34 | 0 (209) | 0 | – |
| Blood tests—renal function (eGFR) and potassium level (number) | 1.06 (201) | 0.34 | 0 (209) | 0 | – |
| Total community-based HCP visits (number) | 0.15 (150) | 0.51 | 0.10 (124) | 0.43 | 0.05 (−0.06 to 0.16) |
| Total hospital contacts (number) | 0.06 (132) | 0.30 | 0.05 (115) | 0.26 | 0.01 (−0.06 to 0.08) |
| All prescription medications (number) | 11.42 (147) | 29.65 | 23.36 (124) | 96.80 | −11.94 (−28.51 to 4.63) |
| Total out-of-pocket items | 3.59 (131) | 5.96 | 4.49 (113) | 6.67 | −0.90 (−2.49 to 0.69) |
| Lost patient work time (number reporting) | 0.00 (186) | 0.00 | 0.02 (191) | 0.144 | −0.02 (−0.04 to -0.00) |
| Lost carer work time (number reporting) | 0.01 (185) | 0.07 | 0.02 (190) | 0.144 | −0.02 (−0.04 to 0.01) |
| **Costs over 24-week period (UK£2021/22):** | | | | | |
| All intervention costs | 122.87 (201) | 13.04 | 0 (209) | 0 | 122.87 (121.09 to 124.64) |
| All community-based HCP costs | 6.28 (150) | 24.83 | 3.75 (124) | 16.46 | 2.53 (−2.60 to 7.66) |
| All hospital contact costs | 7.28 (132) | 36.42 | 5.73 (115) | 28.09 | 1.55 (−6.70 to 9.79) |
| All prescription medication costs | 4.37 (147) | 11.77 | 5.91 (124) | 18.93 | −1.54 (−5.25 to 2.17) |
| **Total costs** | **141.99 (128)** | **57.90** | **15.64 (110)** | **45.62** | **126.35 (112.88 to 139.82)** |
| **Total costs excluding intervention** | **19.61 (128)** | **56.65** | **15.64 (110)** | **45.62** | **3.98 (−9.30 to 17.26)** |
| Total out-of-pocket costs | 69.41 (139) | 113.05 | 82.57 (120) | 148.60 | −13.15 (−45.23 to 18.92) |
| Lost patient and carer productivity | 27.87 (177) | 354.76 | 15.95 (179) | 183.54 | 11.93 (−46.86 to 70.71) |
| **Total costs (wider perspective)** | **252.67 (113)** | **490.19** | **93.53 (100)** | **144.02** | **159.14 (58.86 to 259.41)** |
| **EQ-5D score (CUA)** | | | | | |
| Baseline | 0.887 (200) | 0.148 | 0.860 (209) | 0.200 | 0.027 (−0.008 to 0.061) |
| 6 weeks | 0.894 (176) | 0.135 | 0.863 (179) | 0.168 | 0.031 (−0.001 to 0.063) |
| 12 weeks | 0.904 (174) | 0.138 | 0.877 (166) | 0.177 | 0.027 (−0.007 to 0.061) |
| 24 weeks | 0.909 (163) | 0.153 | 0.890 (136) | 0.180 | 0.019 (−0.019 to 0.057) |
| **Total QALY score over 24 weeks** | **0.417 (162)** | **0.058** | **0.404 (136)** | **0.079** | **0.013 (−0.002 to 0.029)** |
| **Acne-QoL symptom sub-scale score (CEA)** | | | | | |
| Baseline | 13.22 (201) | 4.94 | 12.87 (209) | 4.55 | 0.35 (−0.57 to 1.27) |
| 6 weeks | 16.97 (176) | 5.72 | 15.65 (179) | 5.69 | 1.32 (0.13 to 2.51) |
| 12 weeks | 19.21 (176) | 6.12 | 17.76 (166) | 5.58 | 1.45 (0.20 to 2.69) |
| 24 weeks | 21.22 (163) | 5.86 | 17.39 (136) | 5.80 | 3.83 (2.49 to 5.16) |
| **Change at 24 weeks from baseline** | **8.15 (163)** | **6.12** | **4.46 (136)** | **6.34** | **3.68 (2.26 to 5.11)** |

*Assumes that if spironolactone is found effective it would be prescribed in primary care.
CUA, cost-utility analysis; eGFR, estimated Glomerular Filtration Rate; GP, General Practitioner; HCP, healthcare professional; QALY, quality-adjusted life year.

value is unclear but research comparing spironolactone with oral antibiotics, currently underway,[13] will enable an assessment of plausibility once published.

Of note regarding the wider perspective sensitivity analysis (SA4), the majority of women (97%) reported no impact on their employment as a result of their acne and, thus, it is mainly out-of-pocket expenses driving change from the base-case.

The results of a subgroup analysis undertaken for women aged <25 years and ≥25 years are reported in online supplemental file 3. See online supplemental table S2 found in online supplemental file 3 for results.

## DISCUSSION

This economic study finds a high degree of uncertainty about whether spironolactone is likely to be cost-effective. Our economic evaluation provides a range of estimates for the cost-effectiveness of spironolactone used alongside routine topical treatment. The base-case analysis, where the comparator is no active systemic treatment plus routine topical treatment, and the delivery of the intervention is costed as via primary care, spironolactone was not estimated to be cost-effective in the unadjusted and adjusted complete case analyses. However, in the adjusted analysis, using MI, the ICER was estimated to be just under the £30 000 per QALY threshold. This divergence

**Table 3** Cost-utility analyses and cost-effectiveness analyses results, including sensitivity analyses and subgroup analysis

| CUA analysis (N s, N p) | Incremental cost (95% CI) | Incremental QALYs (95% CI) | ICER | CEAC at £20 000 (£30,000) threshold* |
|---|---|---|---|---|
| Base-case[†], CCA, adjusted (118,101) | 125.36 (111.13 to 139.58) | 0.0019 (−0.0096 to 0.0133) | £67 191 | 23% (35%) |
| Base-case[†], CCA, unadjusted (126,109) | 125.53 (112.15 to 138.91) | 0.0036 (−0.0117 to 0.0189) | £34 770 | 37% (47%) |
| SA1[†], Multiple imputation, adjusted (201,209) | 119.78 (107.99 to 131.57) | 0.0043 (−0.0041 to 0.0127) | £27 879 | 35% (53%) |
| SA2, Secondary care delivery, CCA, adjusted (118,101) | 265.67 (250.52 to 280.82) | 0.0019 (−0.0096 to 0.0133) | £141 955 | 3% (12%) |
| SA3a, oral antibiotic control, CCA, adjusted (118,101) | 17.11 (2.88 to 31.33) | Threshold analysis value[‡]: 0.00057 | | |
| SA3b, oral antibiotic control, MI, adjusted (201, 209) | 11.53 (−0.26 to 23.32) | Threshold analysis value[‡]: 0.00038 | | |
| SA4a, Wider perspective, CCA, adjusted (97,85) | 102.07 (64.21 to 139.92) | −0.0027 (−0.0139 to 0.0085) | Dominated | 9% (15%) |
| SA4b, Wider perspective, MI, adjusted (201,209) | 133.25 (72.52 to 193.93) | 0.0044 (−0.0041 to 0.0129) | £30 249 | 31% (50%) |
| **CEA Analysis (N s, N p)** | **Incremental cost (95% CI)** | **Incremental Acne-QoL symptom (95% CI)** | **Incremental cost per unit change** | – |
| Secondary analysis[†], CCA, adjusted: (119,102) | 126.57 (112.35 to 140.78) | 3.31 (1.90 to 4.72) | £38.21 | – |
| Secondary analysis[†], CCA, unadjusted (127,110) | 126.52 (113.00 to 140.04) | 3.52 (1.94 to 5.11) | £35.91 | – |

*Probability of being cost-effective at the threshold (λ) of £20,000 and £30,000 per QALY. Adjusted analyses, adjusted for stratification variables (centre, baseline severity [IGA<3 vs. ≥3]) and baseline variables (Acne QoL symptom subscale score, use of topical treatments, utility score based on EQ-5D, total costs).
[†]Comparing spironolactone plus routine topical treatment to no active systemic treatment plus routine topical treatment.
[‡]Threshold analysis conducted using a £30,000 threshold, as described in the methods. The value given represents the incremental QALY benefit below which spironolactone compared with oral antibiotic would switch from cost-effective to not cost-effective.
CCA, complete case analysis; CEA, cost-effectiveness Analysis; CEAC, cost-effectiveness acceptability curve; CUA, cost-utility analysis; ICER, incremental cost-effectiveness ratio; N s/N p, number randomised to spironolactone/placebo who were included in the analysis; QALY, quality-adjusted life year.

in conclusion between the complete case and MI analysis demonstrates the impact of missing data (attrition bias) and suggests more weight ought to be placed on the MI analysis.[33] The results of other sensitivity analyses (table 3)

varied around the base-case, adding to the uncertainty of the results.[13]

This economic evaluation followed a HEAP finalised before data were received for analysis, reducing bias in

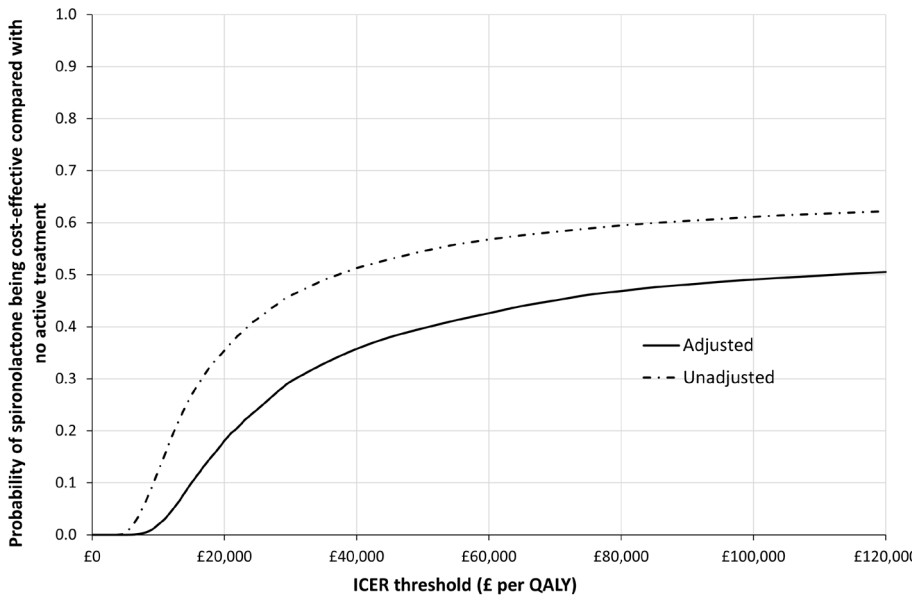

**Figure 2** Cost-effectiveness acceptability curve (CEAC), complete case analysis, adjusted and unadjusted QALYs. ICER, incremental cost-effectiveness ratio; QALYs, quality-adjusted life years.

the results from selective reporting or cherry-picked analyses.[34] Another strength of this economic evaluation is that it can provide reliable estimates of cost-effectiveness based on individual participant-level data, collected at little marginal cost, alongside a randomised controlled trial. This is, however, also a limitation in that within trial, health economic evaluations are constrained by the question, timeframe and data collected, particularly in placebo-controlled trials. In particular, there are five main limitations to acknowledge: (1) the assumptions required to compare spironolactone to inactive systemic treatment; (2) the assumptions required to undertake a sensitivity analysis using oral antibiotics as the comparator; (3) the sensitivity and validity of the EQ-5D-5L in patients with acne; (4) the time frame of the analysis and (5) the use of CCAs rather than the analysis using MI to take account of missing data as the base-case analysis. We look at these in turn below, but all should be borne in mind when interpreting the results.

First, ideally economic evaluations should compare an active treatment to current usual care. The funder for this trial preferred the placebo comparator to current usual care.[17] We wanted our primary analysis to reflect as closely as possible the data collected in the actual trial while reflecting a useful analysis to decision-making in practice. We, therefore, felt the most appropriate comparator would be no active systemic treatment, rather than placebo, which would not reflect reality. Placebos are not used in routine practice, but some evidence of placebo effects has been documented in acne.[5] Therefore, the base-case set out to answer the question of whether spironolactone is cost-effective compared with no active systemic treatment (both groups could use routine topical treatments) to align with the clinical question funded. A limitation of this is that, because it does not account for the potential impact of a placebo effect, it may result in underestimation of the QALY gain with spironolactone compared with not providing spironolactone, and hence, underestimate its cost-effectiveness. We also excluded the research costs associated with administering the placebo (costs of the pills and appointments to administer them) but did include ongoing costs associated with NHS resource use related to acne in both arms of the study. There is also uncertainty about how many, if any, additional GP visits might have occurred in the usual care group if they had actually received usual care as opposed to placebo during the trial. It is not possible to know how costs and effects would differ between our placebo group and a group without any active systemic treatment because we did not have the latter group in the study. We feel the assumptions made are required to make the analysis most useful to practice but acknowledge they may mean the estimates of the cost-effectiveness of spironolactone are conservative.

Second, in practice, clinicians are unlikely to send women away with no active treatment if they consulted with acne persisting beyond 6 months. As advised by the trial clinicians, the clinically important comparator may be another systemic treatment rather than no active systemic treatment. To address this, a sensitivity analysis assuming, for cost purposes, all women in the no active systemic treatment group received an oral antibiotic (in addition to topical treatments) for 24 weeks was planned. This analysis assumed that incremental QALYs remain the same as in the base-case analysis, which we acknowledge is unlikely. There is limited economic evidence comparing oral antibiotics in combination with routine topical treatment compared with routine topical treatment alone.[5] Despite these limitations and while the results of this sensitivity analysis should be interpreted with caution, considering the assumptions made, the analysis serves to provide a lower range estimate for the cost effectiveness of spironolactone that better reflects accepted standard-of-care, based on current NICE guidelines.[32] Further evidence, from randomised controlled trials,[13 14] is required to determine whether this is a likely scenario and to draw conclusions.

Third, the uncertainty highlighted by this study may be impacted, in part, by the method of measuring utility, an area where further research would be valuable. The conclusion reached about cost-effectiveness was sensitive to the estimates of QALYs generated from EQ-5D-5L, despite 46% in the intervention group and 43% in the control group reporting perfect health (EQ-5D-5L health state 11111) at baseline. For these participants, the EQ-5D-5L had no potential to measure improvements in health-related quality of life. This likely contributes to the wide 95% CIs around the incremental QALY estimates in this study, which means we cannot be certain spironolactone improves QALYs rather than have no difference or worsen QALYs. At design stage, there was discussion about the possible use of other instruments; however, the limited published evidence supported the use of the EQ-5D for acne.[35 36] Like Klassen et al,[36] we find that women with persistent acne report most problems on the pain/discomfort and anxiety/depression dimensions of the EQ-5D. Further research using the EQ-5D data generated in this study alongside that elicited in other studies of acne would help inform future studies about the validity and responsiveness of this instrument for acne.

Fourth, we acknowledge that the analysis was conducted for a 24-week timeframe and that were a longer timeframe taken the cost-effectiveness of spironolactone may improve if, for instance, there is a sustained effect once treatment stops. We sought to collect resource use and utility data up to 52 weeks, but due to reduced data completion at 52 weeks (see supplementary material for details), it was not feasible to analyse results to a longer time horizon.[32]

Finally, a CCA was specified in the HEAP as the base-case analysis (with MI as a sensitivity analysis) reflecting a desire to be consistent with the approach undertaken in the Statistical Analysis Plan for the clinical primary outcome. With the benefit of hindsight primary concern ought to have been around the level of missing economic data, which is known to often be greater than that for clinical outcomes. However,

both complete case and MI analyses are reported, as planned, so that the impact of missing data on the results can be clearly seen.

Our study provides estimates of the cost-effectiveness of spironolactone in women with persistent acne using the trial data and a range of scenarios. It highlights that there is considerable uncertainty about whether spironolactone is cost-effective and the need for further research with comparators more akin to clinical practice. The CCA estimated ICERs in excess of the upper NICE threshold of £30 000 per QALY, but this analysis took a conservative approach since it may be that incremental QALYs for spironolactone would have been greater had we been able to control for any placebo effect and had more complete data beyond 24 weeks. When taking into account missing data, the ICER was below the upper NICE threshold, suggesting spironolactone may be considered cost-effective. However, all analyses show a high degree of uncertainty suggestive of a need for further research to allow conclusions to be drawn.

**Author affiliations**
[1]Health Economics Group, Norwich Medical School, University of East Anglia, Norwich, UK
[2]Primary Care Research Centre, School of Primary Care, Population Sciences and Medical Education, University of Southampton, Southampton, UK
[3]Southampton Clinical Trials Unit, University of Southampton and University Hospital Southampton NHS Foundation Trust, Southampton, UK
[4]Centre for Evaluation and Methods Wolfson Institute of Population Health, Queen Mary University of London, London, UK
[5]Centre for Evidence Based Dermatology, University of Nottingham, Nottingham, UK
[6]Public Contributor, Primary Care Research Centre, University of Southampton, Southampton, UK
[7]Population Health Sciences, University of Bristol, Bristol, UK
[8]Skin Research Centre, Hull York Medical School, University of York, York, UK

**Acknowledgements** We would like to thank all PPI contributors, participants, research and clinical staff, the NIHR Clinical Research Network, and the members of the Trial Steering Committee and Data Monitoring Committee for their support. The study was developed with support from the UK Dermatology Clinical Trials Network (UK DCTN). The UK DCTN is grateful to the British Association of Dermatologists and the University of Nottingham for financial support of the Network. The University of Southampton was the research sponsor for this trial.

**Contributors** MS, AML, BS, THS, MJR, NF, KST, PL, JN, GG and IM conceived the study idea and initial study design in response to a NIHR HTA call, with later input from KT, IS, ZE, SR, ML, NVP and SP. All authors contributed to the acquisition of data. Specific advice was given by BS on trial design and medical statistics; and THS on health economic evaluation. Economic analyses were conducted by SP and THS. All authors contributed to the interpretation of data and drafting of this paper, led by SP and THS and approved the final manuscript. Guarantor: THS.

**Funding** This study presents independent research funded by the National Institute for Health and Care Research (NIHR) under its Health Technology Assessment programme (16/13/02). The views expressed are those of the authors and not necessarily those of the NHS, the NIHR or the Department of Health and Social Care. This trial was registered prospectively with the ISRCTN registry (ISRCTN12892056) and EudraCT (2018-003630-33).

**Competing interests** We declare no support from any organisation other than the NIHR for the submitted work; no financial relationships with any organisations that might have an interest in the submitted work; no other relationships or activities that could appear to have influenced the submitted work. LH has received consultancy fees from the University of Oxford on an educational grant funded by Pfizer, unrelated to the submitted work. THS was a member of NIHR HTA Efficient Study Designs-2, HTA Efficient Study Designs Board, HTA End of Life Care and Add-on-Studies, HTA Primary Care Themed Call Board and the HTA Commissioning Board between 2013 to December 2019. She is a steering committee member of the UK Dermatology Clinical Trials Network and Chair of the NIHR Research for Patient Benefit Regional Advisory Panel for the East of England. THS had no part in the decision-making for funding this study.

**Patient and public involvement** Patients and/or the public were involved in the design, or conduct, or reporting, or dissemination plans of this research. Refer to the Methods section for further details.

**Patient consent for publication** Not applicable.

**Ethics approval** Ethical approval for the trial was given by Wales Research Ethics Committee (REC) 3 in January 2019 (18/WA/0420). Participants gave informed consent to participate in the study before taking part.

**Provenance and peer review** Not commissioned; externally peer reviewed.

**Data availability statement** Data may be obtained from a third party and are not publicly available. Consent was not obtained from participants for data sharing but authors will consider reasonable requests to make relevant anonymised participant level data available via the Southampton Clinical Trials Unit Data Sharing Committee.

**ORCID iDs**
Sarah Pyne http://orcid.org/0000-0003-0093-9125
Tracey H Sach http://orcid.org/0000-0002-8098-9220
Beth Stuart http://orcid.org/0000-0001-5432-7437
Kim S Thomas http://orcid.org/0000-0001-7785-7465
Nick Francis http://orcid.org/0000-0001-8939-7312
Matthew J Ridd http://orcid.org/0000-0002-7954-8823
Paul Little http://orcid.org/0000-0003-3664-1873
Ingrid Muller http://orcid.org/0000-0001-9341-6133
Miriam Santer http://orcid.org/0000-0001-7264-5260

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
