## [Reviewer comments · BMJ Open]

ARTICLE DETAILS

TITLE (PROVISIONAL)	Cost-effectiveness of Spironolactone for Adult Female Acne (SAFA): Economic evaluation alongside a randomised controlled trial
AUTHORS	Pyne, Sarah; Sach, Tracey; Lawrence, Megan; Renz, Susanne; Eminton, Zina; Stuart, Beth; Thomas, Kim; Francis, Nick; Soulsby, Irene; Thomas, Karen; Permyakova, Natalia; Ridd, Matthew; Little, Paul; Muller, Ingrid; Nuttall, Jacqui; Griffiths, Gareth; Layton, Alison M; Santer, Miriam

VERSION 1 – REVIEW

REVIEWER	Mavranezouli, Ifigeneia National Collaborating Centre for Mental Health, Research Department of Clinical, Educational and Health Psychology, University College London
REVIEW RETURNED	14-Mar-2023

GENERAL COMMENTS	This is a well-structured, clearly written manuscript, reporting on a study that aimed to assess the cost effectiveness of oral spironolactone relative to no active systematic treatment for women with persistent acne. The economic analysis was conducted alongside a placebo-controlled RCT. Overall, the methodology used for the economic analysis is appropriate and of good quality. I have a number of comments and suggestions, primarily on (1) the assumptions made by the authors regarding the intervention costs of the comparator; (2) the time horizon of the analysis; (3) the use of completer analysis, and not the multiple imputation analysis, as the primary analysis; (4) the appropriateness and assumptions around the use of oral antibiotics as the comparator in a sensitivity analysis; and (5) the conclusion of the analysis regarding the cost-effectiveness of oral spironolactone. I believe that if these can be addressed, the validity and the conclusions of the analysis may be considerably strengthened. 1. Estimation of intervention costs of the comparator: The authors state “No intervention costs (placebo tablets, GP visits or blood tests) were included for the no active systemic treatment group as these would not occur if no intervention was being given (the comparator for this economic evaluation).” So, the ‘no active systematic treatment’ control resource use practically represents ‘no treatment’ (rather than placebo, which was the actual control). However, people in the placebo arm of the trial did presumably have appointments with a healthcare practitioner as part of the intervention, so some of the placebo effect might be attributable to those contacts. Since both the placebo effect and the resource use associated with placebo are in reality higher than those of ‘no
---

	treatment', I think that using the incremental effect of spironolactone vs placebo (which is known to have a systematically higher effect compared with no treatment) and, at the same time, using the incremental cost of spironolactone vs, effectively, no treatment, in order to estimate the cost-effectiveness of oral spironolactone, is biasing results against spironolactone (as it overestimates its incremental cost). Perhaps, in the context of the trial, the intervention cost of placebo should include GP visits, or any other regular visits that were part of the placebo arm 'intervention'. 2. Time horizon of the analysis: I think that one factor contributing to the lack of cost-effectiveness of spironolactone versus placebo is possibly the relatively limited time horizon of the analysis (24 weeks) – if benefits from spironolactone are (even partly) sustained and enjoyed beyond treatment endpoint, with intervention costs being spread over a longer time period, then the cost-effectiveness of spironolactone is likely higher than what is reported in the study. It might be useful to extend the time horizon of the economic analysis, and estimate (probably through extrapolation) total costs and QALYs over 52 weeks. I note that EQ-5D data should be available at that time point according to the study protocol (by looking at the secondary outcomes in the clinical paper). If not, would it be possible to make an estimate of the incremental utility scores at 52 weeks, potentially using the change in incremental QoL data between treatment endpoint and 52 weeks? (I see these are available from Figure 2 of the clinical paper). Depending on data availability (or unavailability!), this approach might only be appropriate as a sensitivity analysis, but it would give a better indication of the cost-effectiveness of spironolactone. 3. I believe that intention-to-treat analysis should be preferred over completer analysis (as also recommended in the Cochrane handbook). Therefore, I would tend to consider the multiple imputation analysis as the base-case analysis (i.e. the main analysis from which conclusions should be drawn), rather than the complete case analysis – but I understand this may depend on the study protocol. An example of the problem in interpretation, when using completer data: See Supplementary Table 3 – CCA analyses. I think the very wide range of the ICERs, determined by available completer data, suggests that CCA analyses should not be trusted. The sub-group analysis of <25 years, CCA, adjusted is based on just 57 women and shows an ICER of more than £263K/QALY. Then the subgroup analysis of ≥25 years, CCA, adjusted, based on 162 women, shows an ICER just below £20K/QALY. I'm not sure I would rely on these findings to conclude that the intervention is cost-effective in women ≥25 years but not cost-effective in younger women. A more extreme example is when a wider perspective, CCA, adjusted is considered, based on 182 women: in this case spironolactone is dominated as it shows a lower effect compared with no treatment; however, the incremental effectiveness of the intervention should not be normally affected by the use of a wider perspective, unless the women providing resource use data relevant to the wider perspective are different from the rest study sample (obviously these results are also affected by the small and non-significant effect of spironolactone vs placebo regarding EQ-5D ratings).
--	--

4. Appropriateness and assumptions around the use of oral antibiotics as the comparator in a sensitivity analysis: The authors did a sensitivity analysis assuming that women in the placebo group took oral antibiotics, stating that participants entering the trial required systemic treatment, therefore making standard of care (oral antibiotics) a 'more appropriate' comparator. However, according to the study inclusion criteria, the main criterion that warranted treatment with oral antibiotics was "acne of sufficient severity" (another criterion is a duration of acne of at least 6 months, but acne may not have been properly and consistently treated over this period, so this is not the main criterion). I note that NICE also recommends two different topical treatment combinations for moderate-to-severe acne (see related comment below), therefore 'placebo plus topical treatment' might also, broadly, comprise standard of care for this population.

Regarding the use of oral antibiotics in the analysis, the authors state: "We assumed that the health-related quality of life remained unchanged from what was collected in the SAFA trial, even though incremental QALYs are likely to be lower with effective antibiotic use.": It is a VERY strong assumption to assume that the QALYs gained following treatment with oral antibiotics would be equal to the QALYs gained following treatment with placebo; effectively, this sensitivity analysis has assumed that oral antibiotics have equal effectiveness with placebo (i.e. that they are ineffective). However, oral antibiotics must be more effective than placebo – otherwise they wouldn't comprise 'standard of care' for this population (nevertheless, it is acknowledged that they may result in side effects in some women, which would lead to some decrement in utility).

The authors report an ICER of £9,169/QALY of spironolactone vs oral antibiotics, which is misleading, as it is estimated based on the assumption that oral antibiotics have the same effect as placebo regarding HRQoL. The authors did report the results of a threshold analysis to ascertain what level of incremental QALYs would switch the intervention between cost-effective and cost-ineffective, and found that incremental QALYs would have to decrease to 0.00057 in order to switch the ICER from cost-effective to cost-ineffective at a £30,000 threshold. The authors have not discussed whether this is a plausible value. The incremental QALYs between spironolactone and placebo are 0.0019 in the base-case adjusted completer analysis. So, in order for incremental QALYs to decrease to 0.00057 when replacing placebo with oral antibiotics, the latter should have an incremental QALY of 0.00133 versus placebo, which looks realistic (it's broadly similar to the incremental QALYs of spironolactone vs placebo). So, threshold analysis is not useful in determining whether spironolactone can be safely considered cost-effective compared with oral antibiotics – in fact, it suggests that there is high uncertainty in the estimated ICER between the two.

I suggest the authors identify a meta-analysis of RCTs or individual RCTs comparing oral antibiotics with placebo, ideally reporting EQ-5D ratings (or, if these are not available, other QoL ratings), in order to estimate the relative effects between oral antibiotics and placebo, and apply these relative effects (probably with some transformation and assumptions) onto the placebo effect obtained from their RCT, in order to estimate more accurately the incremental QALYs and, ultimately, the relative

	cost-effectiveness between spironolactone and oral antibiotics. The current sensitivity analysis (assuming oral antibiotic cost and placebo effect in the control arm) lacks validity and should not be used to draw any conclusions, however rough, on the relative cost-effectiveness between oral spironolactone and oral antibiotics. 5. Conclusion of the analysis regarding the cost-effectiveness of spironolactone: The authors conclude that, compared to placebo, spironolactone may not be cost-effective (at best it was marginally cost-effective using NICE criteria), but when using oral antibiotics as a more appropriate comparator, then spironolactone appears cost-effective and avoids wider societal costs of over-use of antibiotics. Even after ignoring point 4 above, I disagree with this interpretation. If spironolactone is not cost-effective vs placebo, but is cost-effective vs oral antibiotics, this suggests that oral antibiotics are also not cost-effective vs placebo. So, the overall conclusion is that neither spironolactone nor antibiotics should be used for the treatment of acne in this population; it is not that spironolactone should be used instead of oral antibiotics. As I suggested earlier, and if we assume that the current conclusions of the economic analysis are valid, suggesting that oral spironolactone and, indirectly, oral antibiotics are not cost-effective vs no systematic treatment, it may be more appropriate for this population to be treated with a combination of topical treatments, as recommended by NICE. On the other hand, considering the cost-effectiveness of oral antibiotics, it is noted that, according to the NICE acne guideline economic analysis, oral antibiotics combined with topical treatments are among the most cost-effective treatment options for people with moderate-to-severe acne. Other comments: Introduction 6. The authors state that “whilst a fixed combination topical agent plus oral lymecycline or doxycycline once daily is recommended for moderate-to-severe acne”: actually, in addition to the above, the NICE guideline also recommends the following for moderate-to-severe acne:  • a fixed combination of topical adapalene with topical benzoyl peroxide (this is recommended for any acne severity) • a fixed combination of topical tretinoin with topical clindamycin (this is also recommended for any acne severity) • topical azelaic acid with either oral lymecycline or oral doxycycline. So, a population with moderate-to-severe acne does not necessarily need to be treated with systemic therapy. 7. Consider adding the perspective of the analysis when stating the objective of the study, for clarity. Also, it would be good to clarify that it is the British NHS perspective (also in the abstract). Patients and methods 8. Is contraceptive counselling necessary before initiation of spironolactone? If so, has it been included in the cost as part of the GP/dermatology nurse initial visit? Please clarify. Methods - Measuring costs
--	---

	9 “The intervention was costed as described in Figure 1, which assumes that standard care, if adopted, will be delivered in primary care, including two GP visits (unless >45 years of age), baseline blood test and the cost of spironolactone (50 mg 6 weeks, 100 mg 18 weeks).” I presume authors mean standard treatment with spironolactone when they refer to ‘standard care’ – please rephrase. Methods - Economic analysis 10. What was the usefulness of the secondary analysis that estimated incremental cost per unit change on the Acne-QoL symptom subscale? As the authors acknowledge, it is not possible to make conclusions on cost-effectiveness using this output (unlike cost/QALY, where the NICE cost-effectiveness threshold, or other country empirical thresholds can be used to draw conclusions on cost-effectiveness). Was it used because it was part of the protocol for the economic analysis? Results – participant characteristics 11. “Of the 410 women recruited to the trial, 201 were randomly assigned to spironolactone alongside routine topical treatment and 209 allocated to placebo alongside routine topical treatment at the start of the trial.”: I think this statement is not accurate – routine topical treatment, although allowed during the study, was not part of the protocol, and only 60% of women received topical treatment (see below). This should be clarified. 12. “topical acne treatments were used by 83% (340/410)” – I think this may not be correct. According to Table S1 of the clinical study, 340 women responded yes to the question “Have you used, or are you currently using, topical treatments?” So, a positive answer does not mean that the women were taking topical treatments during the trial (they could have used before the trial). Moreover, the next question in that table, asking for specific treatments they were taking, is for women who responded ‘now’ or ‘now and in the past’ – so by adding the numbers reported for this answer, it looks like around 60% of women were using topical acne treatments during the trial. 13. Table 2: Were incremental utility scores controlled for baseline? Women in the spironolactone arm have higher utility scores at baseline compared with those in the placebo arm. Notably, the difference in EQ-5D scores at baseline is the second largest difference in utility scores observed between the two arms, following the difference in utility scores at 8 weeks. 14. Sensitivity analyses – Figure S1 shows the intervention resource use as delivered via secondary care per trial protocol, which includes blood tests only at baseline. On the other hand, Figure 1 shows blood tests being undertaken also at 10 and 22 weeks in primary care, although it is noted that only a proportion of people would need these. Does this discrepancy between Figure 1 and Figure S1 regarding these tests reflect only a difference in costing between the two scenarios that is specific to this analysis, or also a difference in routine clinical practice between primary and secondary care?
--	--

REVIEWER	Borre , Ethan Duke University
REVIEW RETURNED	02-Apr-2023

GENERAL COMMENTS

The authors report a cost-effectiveness analysis of the SAFA trial, in which 410 women were randomized to receive spironolactone vs. placebo for 24-weeks. The authors collected detailed cost and quality of life data to inform the CEA. This study has several strengths including its treatment of uncertainty and inclusion of extensive sensitivity analysis to address missing data and lack of standard of care comparator. That said, I have some suggestions to improve the analysis. In particular, I would recommend the authors amend their conclusions to better reflect the uncertainty underlying their cost-effectiveness results (see last comment below).

Abstract

- In reporting results, the authors mention adjusted and unadjusted results, but it is unclear how these results were adjusted in the Abstract methods section.
- Please report what the NICE threshold value is.
- NICE acronym is not defined (may be helpful for BMJ Open global audience).
- Please do not use the term “cost-ineffective,” rather the intervention was not cost-effective under the NICE willingness-to-pay.

Methods

- Please include the surveys used to collect patient costs as an appendix.
- Is assuming no intervention costs for the no treatment group a fair comparison? These people would likely present for care and receive other treatment. I would assume that a proportion would likely be seen by a GP or dermatologist for severe acne even if they opt to not use treatment? This could be a sensitivity analysis.
- “Several sensitivity analyses were agreed before...” Should include agreed upon.
- In the SA comparing spiro to systemic antibiotics, why assume that the systemic antibiotics had QALY effects similar to placebo in the SAFA trial? I would recommend using literature estimates on the effects of systemic antibiotics on acne clearance, or even better HRQoL utility if available. This can then better be compared to spiro, even if the underlying population is different.
- Is it appropriate to assume the differential missingness was happening at random? One could imagine that those in the placebo group were dissatisfied with their acne treatment and therefore dropped out of the study/did not complete study forms.
- Week 12 patients could take hormonal treatment, oral antibiotics, or isotretinoin
- Have the authors considered completing a CHEERS checklist?

Results

- Please clarify the time horizon for the ~0.4 estimated QALYs – assuming it is 6 months.
- The authors should note that the 95% CI for the QALY difference between groups included 0. This means that the spiro may impart no difference, or even worse HRQoL utility compared to placebo. I understand this uncertainty is partially accounted for in the PSA, but the authors should highlight this fact in the Abstract, Results, and Discussion.
- Please correct the reference error on page 8.
- It is important that the authors compared spironolactone treatment to “standard of care” oral antibiotics as these would

	likely be the alternative clinical strategies. The authors should clarify the QALY benefit assumed for oral antibiotics. If it was identical to placebo in the SAFA trial, this is not an adequate comparison. As suggested above, a more appropriate comparison might be to obtain QALY effect estimates from the literature or other sources and compare that to what was estimated for spironolactone. Discussion  • “where a more realistic comparator (standard of care based upon NICE guidelines) is used, spironolactone is found to be highly cost-effective” Please clarify the QALY assumptions for SOC treatment and adjust this statement. • “On balance, when dealing with missing data or considering a more realistic comparator (providing antibiotic treatment), the likely conclusion is that spironolactone is cost-effective.” I am not convinced by the results that spironolactone is likely cost-effective. The trial did not find a significant difference in QALY effects, and the base-case analysis found spironolactone to not be cost-effective at standard NICE WTP. When including uncertainty, spironolactone is cost-effective in only ~30% of simulations at standard NICE WTP. The authors should base their conclusions on the results of their basecase analysis, and better address the underlying uncertainty. There is likely high value of future information clarifying this uncertainty, in particular comparing spironolactone to oral antibiotics, and future research might better elucidate the cost-effectiveness of spiro for acne treatment.
--	---

VERSION 1 – AUTHOR RESPONSE

Reviewer 1	
Comments to the Author: This is a well-structured, clearly written manuscript, reporting on a study that aimed to assess the cost effectiveness of oral spironolactone relative to no active systematic treatment for women with persistent acne. The economic analysis was conducted alongside a placebo-controlled RCT. Overall, the methodology used for the economic analysis is appropriate and of good quality. I have a number of comments and suggestions, primarily on (1) the assumptions made by the authors regarding the intervention costs of the comparator; (2) the time horizon of the analysis; (3) the use of completer analysis, and not the multiple imputation analysis, as the primary analysis; (4) the appropriateness and assumptions around the use of oral antibiotics as the comparator in a sensitivity analysis; and	We thank the reviewer for their useful comments on the paper. We are pleased the reviewer identifies that we choose to compare oral spironolactone to no active systemic treatment in our analysis.

(5) the conclusion of the analysis regarding the cost-effectiveness of oral spironolactone. I believe that if these can be addressed, the validity and the conclusions of the analysis may be considerably strengthened.	
1. Estimation of intervention costs of the comparator: The authors state “No intervention costs (placebo tablets, GP visits or blood tests) were included for the no active systemic treatment group as these would not occur if no intervention was being given (the comparator for this economic evaluation).” So, the ‘no active systemic treatment’ control resource use practically represents ‘no treatment’ (rather than placebo, which was the actual control). However, people in the placebo arm of the trial did presumably have appointments with a healthcare practitioner as part of the intervention, so some of the placebo effect might be attributable to those contacts. Since both the placebo effect and the resource use associated with placebo are in reality higher than those of ‘no treatment’, I think that using the incremental effect of spironolactone vs placebo (which is known to have a systematically higher effect compared with no treatment) and, at the same time, using the incremental cost of spironolactone vs, effectively, no treatment, in order to estimate the cost-effectiveness of oral spironolactone, is biasing results against spironolactone (as it overestimates its incremental cost). Perhaps, in the context of the trial, the intervention cost of placebo should include GP visits, or any other regular visits that were part of the placebo arm ‘intervention’.	Ideally economic evaluations should compare an active treatment to current usual care. The funder for this trial preferred the placebo comparator to current usual care. We wanted our primary analysis to reflect as closely as possible the data collected in the actual trial whilst reflecting a useful analysis to decision making in practice. We therefore felt the most appropriate comparator would be no active systemic treatment as placebo would not reflect reality. (Michael Drummond and Mark Sculpher. Common Methodological Flaws in Economic Evaluations. Medical Care, Jul., 2005, Vol. 43, No. 7, Supplement: Cost-Effectiveness Analysis in US Healthcare Decision-Making Where Is It Going? (Jul., 2005), pp. II5-II14). We did include resource use and costs related to the participants acne in both groups of the study. It is only the costs associated with giving the placebo (i.e. the placebo tablets, visits to the GP to prescribe the tablets) that were not included as these would not occur in practice, that is they represent research costs. We discussed the limitation of undertaking an economic evaluation alongside a placebo-controlled trial and the impact of placebo effects in the discussion section so feel we appropriately acknowledge this important limitation. To include the cost of the placebo tablets and associated placebo intervention costs would change our comparator to one that is less relevant to decision makers. We acknowledge the analysis is conservative against spironolactone. The results we present provide a range on the estimates of cost-effectiveness which helps decision makers understand the uncertainty associated with the decision about whether to fund spironolactone or not. We have revised the paper to make it clearer in the method section that acne-related costs are included for both arms and have referred to ‘no active systemic treatment’ as the comparator instead of ‘placebo’ unless referring to the clinical trial itself. See Page 5: “No intervention costs (placebo tablets, GP visits to prescribe placebo tablets or blood tests) were included for the no active systemic treatment group as these would not occur if no intervention was being given (the comparator for this economic evaluation).”

2. Time horizon of the analysis: I think that one factor contributing to the lack of cost-effectiveness of spironolactone versus placebo is possibly the relatively limited time horizon of the analysis (24 weeks) – if benefits from spironolactone are (even partly) sustained and enjoyed beyond treatment endpoint, with intervention costs being spread over a longer time period, then the cost-effectiveness of spironolactone is likely higher than what is reported in the study. It might be useful to extend the time horizon of the economic analysis, and estimate (probably through extrapolation) total costs and QALYs over 52 weeks. I note that EQ-5D data should be available at that time point according to the study protocol (by looking at the secondary outcomes in the clinical paper). If not, would it be possible to make an estimate of the incremental utility scores at 52 weeks, potentially using the change in incremental QoL data between treatment endpoint and 52 weeks? (I see these are available from Figure 2 of the clinical paper). Depending on data availability (or unavailability!), this approach might only be appropriate as a sensitivity analysis, but it would give a better indication of the cost-effectiveness of spironolactone.	Our original plan was to undertake extrapolation to 52 weeks using data collected from the observation phase of the study. However, unfortunately data completion rates were poor at 52 weeks with around 93% missing cost data and around 58% missing utility data. In view of this we do not believe undertaking extrapolation would be appropriate, but we have added a table showing Proportion of Missing values for key variables at the different time points by group (see Supplementary Table S4) and mean (SD) resource use/costs and utility for available case data at 52 weeks in supplementary materials, along with some explanation (see Supplementary Table S5). We have added acknowledgment of this potential limitation in the discussion section: “Fourthly, we acknowledge that the analysis was conducted for a 24-week timeframe and that were a longer timeframe taken the cost-effectiveness of spironolactone may improve if, for instance, there is a sustained effect once treatment stops. We sought to collect resource use and utility data up to 52 weeks but due to reduced data completion at 52 weeks (see supplementary Tables S4 and S5) it was not feasible to analyse results to a longer time horizon. It is also the case that this study is likely to underestimate the economic benefit of spironolactone as, given the timeframe, it was not feasible to capture the potential benefits of reduced prescribing of antibiotics on antimicrobial resistance.”
3. I believe that intention-to-treat analysis should be preferred over completer analysis (as also recommended in the Cochrane handbook). Therefore, I would tend to consider the multiple imputation analysis as the base-case analysis (i.e. the main analysis from which conclusions should be drawn), rather than the complete case analysis – but I understand this may depend on the study protocol. An example of the problem in interpretation, when using completer data: See Supplementary Table 3 – CCA analyses. I think the very wide range of the ICERs, determined by available completer data, suggests that CCA analyses should not be trusted. The sub-group analysis of <25 years,	We agree with the reviewer and in hindsight wish we had specified the multiple imputation analysis as the base case analysis, but our Health Economics Analysis Plan specified complete case analysis as the base case analysis to be in keeping with the primary analysis in the Statistical Analysis Plan and protocol such that it would be inappropriate to change it at this stage. We have added the following text to the limitations section of the discussion: “Finally, a complete case analysis was specified in the Health Economic Analysis Plan as the base case analysis (with multiple imputation as a sensitivity analysis) reflecting a desire to be consistent with the approach undertaken in the Statistical Analysis Plan for the clinical primary outcome. With the benefit of hindsight primary concern ought to have been around the level of missing economic data, which is known to often be greater than that for clinical

CCA, adjusted is based on just 57 women and shows an ICER of more than £263K/QALY. Then the subgroup analysis of ≥25 years, CCA, adjusted, based on 162 women, shows an ICER just below £20K/QALY. I'm not sure I would rely on these findings to conclude that the intervention is cost-effective in women ≥25 years but not cost-effective in younger women. A more extreme example is when a wider perspective, CCA, adjusted is considered, based on 182 women: in this case spironolactone is dominated as it shows a lower effect compared with no treatment; however, the incremental effectiveness of the intervention should not be normally affected by the use of a wider perspective, unless the women providing resource use data relevant to the wider perspective are different from the rest study sample (obviously these results are also affected by the small and non-significant effect of spironolactone vs placebo regarding EQ-5D ratings).	outcomes. However, both complete case and multiple imputation analyses are reported, as planned, so that the impact of missing data on the results can be clearly seen." (see page 12 of the manuscript with tracked changes). Part of the reason for not presenting some of the results in Table S3 in the main paper was due to the high proportion of missing data. In reporting these results in the supplementary material we acknowledge this limitation and influence on the results. With regards to the sub-group analysis, we state: "Whilst this finding is in line with the clinical findings, it ought to be interpreted with caution given the small sample sizes necessitated by splitting the dataset into subgroups" and have added "combined with missing data". (See Page 2 of supplementary material under the heading 'Subgroup analysis by age').
4. Appropriateness and assumptions around the use of oral antibiotics as the comparator in a sensitivity analysis: The authors did a sensitivity analysis assuming that women in the placebo group took oral antibiotics, stating that participants entering the trial required systemic treatment, therefore making standard of care (oral antibiotics) a 'more appropriate' comparator. However, according to the study inclusion criteria, the main criterion that warranted treatment with oral antibiotics was "acne of sufficient severity" (another criterion is a duration of acne of at least 6 months, but acne may not have been properly and consistently treated over this period, so this is not the main criterion). I note that NICE also recommends two different topical treatment combinations for moderate-to-severe acne (see related comment below), therefore 'placebo plus topical treatment' might also, broadly, comprise standard of care for this population.	While we agree that topical treatment combinations are amongst the treatment options available to patients with moderate to severe acne, the inclusion criteria for the patients in this trial included: "acne of sufficient severity to warrant treatment with oral antibiotics, as judged by the trial clinician...". This, combined with expert opinion that said they were unlikely to send these women away without further treatment, is the justification for presenting this sensitivity analysis, which is aimed to address the decision clinicians are likely to be making in reality. We do acknowledge though that this is only an estimate within the limits of the trial data available (see second point of the discussion on page 11 of the main paper with tracked changes). Both reviewers make a similar suggestion in regard to sourcing estimates of utility from meta-analysis of RCTs or individual RCTs comparing oral antibiotics with placebo for our sensitivity analysis comparing to standard of care (oral antibiotics). However, we were unable to find published estimates of EQ-5D or utility, including for moderate to severe acne. We were able to find a systematic review and network meta-analysis of topical pharmacological, oral pharmacological, physical and combined treatments for acne vulgaris but this does not report EQ-5D ratings or other QOL ratings rather

Regarding the use of oral antibiotics in the analysis, the authors state: “We assumed that the health-related quality of life remained unchanged from what was collected in the SAFA trial, even though incremental QALYs are likely to be lower with effective antibiotic use.”: It is a VERY strong assumption to assume that the QALYs gained following treatment with oral antibiotics would be equal to the QALYs gained following treatment with placebo; effectively, this sensitivity analysis has assumed that oral antibiotics have equal effectiveness with placebo (i.e. that they are ineffective). However, oral antibiotics must be more effective than placebo – otherwise they wouldn’t comprise ‘standard of care’ for this population (nevertheless, it is acknowledged that they may result in side effects in some women, which would lead to some decrement in utility). The authors report an ICER of £9,169/QALY of spironolactone vs oral antibiotics, which is misleading, as it is estimated based on the assumption that oral antibiotics have the same effect as placebo regarding HRQoL. The authors did report the results of a threshold analysis to ascertain what level of incremental QALYs would switch the intervention between cost-effective and cost-ineffective, and found that incremental QALYs would have to decrease to 0.00057 in order to switch the ICER from cost-effective to cost-ineffective at a £30,000 threshold. The authors have not discussed whether this is a plausible value. The incremental QALYs between spironolactone and placebo are 0.0019 in the base-case adjusted completer analysis. So, in order for incremental QALYs to decrease to 0.00057 when replacing placebo with oral antibiotics, the latter should have an incremental QALY of 0.00133 versus placebo, which looks realistic (it’s broadly similar to the incremental QALYs of spironolactone vs placebo). So, threshold analysis is not	they look at “percentage change in total lesion count from baseline, treatment discontinuation for any reason, and discontinuation owing to side-effects”. They report that “The quality of included RCTs was moderate to very low, with evidence of inconsistency between direct and indirect evidence” (Mavranouzouli I, Daly CH, Welton NJ, et al. A systematic review and network meta-analysis of topical pharmacological, oral pharmacological, physical and combined treatments for acne vulgaris. Br J Dermatol. 2022 Nov;187(5):639-649). We have investigated the possibility of using assumptions around utility values from the model used in the paper by the same authors: “Cost-effectiveness of topical pharmacological, oral pharmacological, physical and combined treatments for acne vulgaris”. However, this was not possible as utility values for specific treatment regimens were not used, rather values related to different perceived improvements were combined with the results from the aforementioned network meta-analysis to give an estimate. Further these utility estimates were based on two studies one a small study of acne patients in England with mild to moderate acne (Klassen AF, Newton JN, Mallon E. Measuring quality of life in people referred for specialist care of acne: comparing generic and disease-specific measures. J Am Acad Dermatol 2000; 43 (2 Pt 1):229-33) and the other a study undertaken in Saudi-Arabia also in mild to moderate cases (Al Robaee AA. Assessment of general health and quality of life in patients with acne using a validated generic questionnaire. Acta Dermatovenerol Alp Pannonica Adriat 2009; 18:157-64). The estimates of utility from these two studies are not relevant to our study population (the participants had mild to moderate acne and were on a mix of treatments with results not presented separately for different types of treatment), the studies were small and did not have a controlled design. The reviewer states that “effectively, this sensitivity analysis has assumed that oral antibiotics have equal effectiveness with placebo (i.e. that they are ineffective)”, but in fact we are assuming that oral antibiotics plus routine topical treatments have equal effectiveness to placebo plus routine topical treatment. Given the potential for placebo effects (discussed previously) and evidence from the previously mentioned network meta-analysis, we feel this is not an unreasonable assumption, given that we are explicit about the limitations. Further review of the network meta-analysis shows that in moderate-to-severe acne
--	---

useful in determining whether spironolactone can be safely considered cost-effective compared with oral antibiotics – in fact, it suggests that there is high uncertainty in the estimated ICER between the two.

I suggest the authors identify a meta-analysis of RCTs or individual RCTs comparing oral antibiotics with placebo, ideally reporting EQ-5D ratings (or, if these are not available, other QoL ratings), in order to estimate the relative effects between oral antibiotics and placebo, and apply these relative effects (probably with some transformation and assumptions) onto the placebo effect obtained from their RCT, in order to estimate more accurately the incremental QALYs and, ultimately, the relative cost-effectiveness between spironolactone and oral antibiotics. The current sensitivity analysis (assuming oral antibiotic cost and placebo effect in the control arm) lacks validity and should not be used to draw any conclusions, however rough, on the relative cost-effectiveness between oral spironolactone and oral antibiotics.

topical treatment only (with lincosamide and retinoid) demonstrated 44.43% (95% CI 29.2–60.2) change in acne lesions, while oral antibiotic with topical treatment (oral tetracycline + benzoyl peroxide + topical retinoid) was 43.53% (29.49–57.70). Other topical combinations and oral tetracycline alone appeared less effective than these, but oral tetracycline alone or in combination with topical treatments did not demonstrate superiority over topical treatment combinations alone. Therefore, we believe our assumption, for this sensitivity analysis, that an oral antibiotic plus routine topical treatment may not result in additional QALY gains beyond those achieved by taking a placebo alongside routine topical treatment is not unreasonable.

We acknowledge in the paper that assuming incremental QALYs remain the same as when compared to placebo plus routine topical treatment is a strong assumption which is why we conducted a threshold analysis to see at what point spironolactone would switch from being cost-effective to not being cost-effective in this analysis by varying the level of incremental QALYs. The reviewer extrapolates that “in order for incremental QALYs to decrease to 0.00057 when replacing placebo with oral antibiotics, the latter should have an incremental QALY of 0.00133 versus placebo”, when in fact this would be versus topical treatment alone, which looks realistic for oral antibiotics vs. nothing, but not for oral antibiotics + topical treatments vs. topical treatments alone (the relevant comparison here). We, therefore, feel this analysis is suitably qualified and provides a range on the cost effectiveness estimates which can be interpreted sensibly given the caveats and threshold analysis reported. We have however added some text to discuss the threshold value and that this demonstrates high uncertainty which would likely benefit from further research. We reference an ongoing trial which aims to fill this gap in the evidence (Poinas A, Lemoigne M, le Naour S, et al. FASCE, the benefit of spironolactone for treating acne in women: study protocol for a randomized double-blind trial. *Trials* 2020; 21: 571). See first and third paragraphs on Page 10 of the main paper with tracked changes, including:

“Threshold analysis around the level of incremental QALYs assumed in this analysis, demonstrates that the conclusion is likely to hold true even if the incremental QALYs gained in the spironolactone group compared with the oral antibiotic group are decreased to 0.00057.

	We await further evidence to determine whether this is a likely scenario.” We have amended the wording in the ‘patients and methods’ section to explain the use of this comparator more clearly and in the discussion to clarify the limitations and reasoning for the method we used. See page 5 of the main paper with tracked changes: “Although in the clinical trial, spironolactone plus routine topical treatment was compared to placebo plus routine topical treatment, it is most appropriate in economic evaluations to compare an active treatment to current usual care. Therefore, to utilise the data collected in the trial whilst reflecting a useful analysis to decision makers in practice, this economic evaluation compared spironolactone plus routine topical treatment to not active systemic treatment plus routine topical treatment.”
5. Conclusion of the analysis regarding the cost-effectiveness of spironolactone: The authors conclude that, compared to placebo, spironolactone may not be cost-effective (at best it was marginally cost-effective using NICE criteria), but when using oral antibiotics as a more appropriate comparator, then spironolactone appears cost-effective and avoids wider societal costs of over-use of antibiotics. Even after ignoring point 4 above, I disagree with this interpretation. If spironolactone is not cost-effective vs placebo, but is cost-effective vs oral antibiotics, this suggests that oral antibiotics are also not cost-effective vs placebo. So, the overall conclusion is that neither spironolactone nor antibiotics should be used for the treatment of acne in this population; it is not that spironolactone should be used instead of oral antibiotics. As I suggested earlier, and if we assume that the current conclusions of the economic analysis are valid, suggesting that oral spironolactone and, indirectly, oral antibiotics are not cost-effective vs no systematic treatment, it may be more appropriate for this population to be treated with a combination of topical treatments, as recommended by NICE.	We found that spironolactone plus routine topical treatment has not been shown cost-effective compared to no active systemic treatment plus routine topical treatment, while a sensitivity analysis has shown evidence that spironolactone plus routine topical treatment may be cost-effective compared to oral antibiotics plus routine topical treatment. This holds true on the assumption that oral antibiotics in combination with routine topical treatment do not convey any QALY gains compared with placebo plus routine topical treatment. The reduction in cost difference is what accounts for the reduced ICER and increased cost-effectiveness in this sensitivity analysis. We have added text in a number of places to make clearer the uncertainty our results reveal. For instance, page 12 of the main paper with tracked changes now states: “Our study provides estimates of the cost-effectiveness of spironolactone in women with persistent acne using the trial data and a range of scenarios. It highlights that there is uncertainty about whether spironolactone is cost effective and the need for further research with comparators more akin to clinical practice. The complete case analysis estimated ICERs in excess of the upper NICE threshold of £30,000 per QALY but this analysis took a conservative approach since it may be that incremental QALYs for spironolactone would have been greater had we been able to control for any placebo effect and had more complete data beyond 24 weeks. When taking into account missing data the ICER was below the upper NICE threshold suggesting

On the other hand, considering the cost-effectiveness of oral antibiotics, it is noted that, according to the NICE acne guideline economic analysis, oral antibiotics combined with topical treatments are among the most cost-effective treatment options for people with moderate-to-severe acne.	spironolactone may be considered cost-effective. However, all analyses show a high degree of uncertainty suggestive of a need for further research to reduce the uncertainties. Our study was primarily designed to address the question of whether spironolactone is (cost) effective compared to no active systemic treatment, as this was the trial that was funded. We undertook sensitivity analysis against potential standard of care because this is a more relevant question for clinicians and we acknowledge the limitations of this sensitivity analysis in the paper. Our data cannot be used to answer the questions raised by the reviewer. The situation is also more complicated than represented by the reviewer, our sample were women with persistent acne that had not responded to first line topical treatments. The economic evaluation considered by NICE did not have any evidence available to inform it on the use of spironolactone. Our study is one piece of evidence to add to this topic it does not claim to answer the wider question of how best to manage acne for all people living with acne. We look forward to head-to-head comparisons of spironolactone and doxycycline but in the meantime this study offers useful data.
Other comments:	
Introduction	
6. The authors state that “whilst a fixed combination topical agent plus oral lymecycline or doxycycline once daily is recommended for moderate-to-severe acne”: actually, in addition to the above, the NICE guideline also recommends the following for moderate-to-severe acne:  • a fixed combination of topical adapalene with topical benzoyl peroxide (this is recommended for any acne severity) • a fixed combination of topical tretinoin with topical clindamycin (this is also recommended for any acne severity)  • topical azelaic acid with either oral lymecycline or oral doxycycline. So, a population with moderate-to-severe acne does not necessarily need to be treated with systemic therapy.	As these women had persistent acne (facial acne for at least 6 months) and were considered to have acne of sufficient severity to warrant treatment with oral antibiotics, as judged by the trial clinician (see eligibility criteria, clinical paper), the following guidance from NICE may be more relevant: “If acne fails to respond adequately to a 12-week course of a first-line treatment option and at review the severity is...  - moderate to severe, and the treatment did not include an oral antibiotic: offer another option which includes an oral antibiotic from the table of treatment choices (table 1)” - - It was not possible to make comparisons to each of the treatment recommendations listed for moderate-to-severe acne in the NICE guideline. Given the high prevalence of oral antibiotic prescribing in acne (Francis NA, Entistle K, Santer M et al. The management of acne vulgaris in primary care: a cohort study of consulting and prescribing patterns using the Clinical Practice Research Datalink BJD 2017; 176: 107-15), it seems likely that women who have acne persisting over 6 months are

	likely to be prescribed a regimen that includes oral antibiotic.
7. Consider adding the perspective of the analysis when stating the objective of the study, for clarity. Also, it would be good to clarify that it is the British NHS perspective (also in the abstract).	Amended as suggested.
Patients and methods	
8. Is contraceptive counselling necessary before initiation of spironolactone? If so, has it been included in the cost as part of the GP/dermatology nurse initial visit? Please clarify.	Contraceptive counselling takes place as part of the initial GP visit and would not incur additional resource use over and above this.
Methods - Measuring costs	
9 “The intervention was costed as described in Figure 1, which assumes that standard care, if adopted, will be delivered in primary care, including two GP visits (unless >45 years of age), baseline blood test and the cost of spironolactone (50 mg 6 weeks, 100 mg 18 weeks).” I presume authors mean standard treatment with spironolactone when they refer to ‘standard care’ – please rephrase.	Amended as suggested.
Methods - Economic analysis	
10. What was the usefulness of the secondary analysis that estimated incremental cost per unit change on the Acne-QoL symptom subscale? As the authors acknowledge, it is not possible to make conclusions on cost-effectiveness using this output (unlike cost/QALY, where the NICE cost-effectiveness threshold, or other country empirical thresholds can be used to draw conclusions on cost-effectiveness). Was it used because it was part of the protocol for the economic analysis?	Yes and because future economic studies in acne may undertake such analysis and this could enable future comparisons. Added short explanation to methods section. See page 6 of the main paper with tracked changes: “A secondary economic outcome was the Acne-QoL symptom sub-scale score (five questions with seven responses to each)^{23,24} at week-24, used as an estimate of effectiveness, which enables comparison with future economic studies in acne.”
Results – participant characteristics	
11. “Of the 410 women recruited to the trial, 201 were randomly assigned to spironolactone alongside routine topical treatment and 209 allocated to placebo alongside routine topical treatment at the start of the trial.”: I think this statement is not accurate – routine topical treatment, although allowed during the study, was not part of the	The wording has been amended in the manuscript to make the randomisation clearer. See page 7/8 of the main paper with tracked changes: “Of the 410 women recruited to the trial, 201 were randomly assigned to spironolactone and 209 allocated to placebo at the start of the trial. All were allowed to continue routine topical treatment.”

protocol, and only 60% of women received topical treatment (see below). This should be clarified.	We already report the number of participants who chose to use topical treatments in the paragraph highlighted by the reviewer (“topical acne treatments were used by 83% (340/410)”).
12. “topical acne treatments were used by 83% (340/410)” – I think this may not be correct. According to Table S2 of the clinical study, 340 women responded yes to the question “Have you used, or are you currently using, topical treatments?” So, a positive answer does not mean that the women were taking topical treatments during the trial (they could have used before the trial). Moreover, the next question in that table, asking for specific treatments they were taking, is for women who responded ‘now’ or ‘now and in the past’ – so by adding the numbers reported for this answer, it looks like around 60% of women were using topical acne treatments during the trial.	We now more clearly state in the paper that this was before or during the trial. See page 8 of the main paper with tracked changes: “...at baseline 83% (340/410) participants were using or had used topical treatments, and the majority (75% [306/410]) had acne for two or more years.” With regards to summing the acne treatments in table S2, there were a proportion of people who did not respond to these questions, so it’s not possible to total the responses to know the proportion that were taking topical treatments.
13. Table 2: Were incremental utility scores controlled for baseline? Women in the spironolactone arm have higher utility scores at baseline compared with those in the placebo arm. Notably, the difference in EQ-5D scores at baseline is the second largest difference in utility scores observed between the two arms, following the difference in utility scores at 8 weeks.	Table 2 is based on available case data and we now more clearly specify this. Baseline utility scores are adjusted for in adjusted incremental analyses.
14. Sensitivity analyses – Figure S1 shows the intervention resource use as delivered via secondary care per trial protocol, which includes blood tests only at baseline. On the other hand, Figure 1 shows blood tests being undertaken also at 10 and 22 weeks in primary care, although it is noted that only a proportion of people would need these. Does this discrepancy between Figure 1 and Figure S1 regarding these tests reflect only a difference in costing between the two scenarios that is specific to this analysis, or also a difference in routine clinical practice between primary and secondary care?	Figure 1 reflects how standard treatment of spironolactone is likely to be delivered in practice, i.e. in primary care and in line with existing guidelines (see statement and references in ‘patient and methods’/‘measuring costs’) and outlines how the intervention was costed for the base-case analysis. Figure S1 reflects the study protocol, i.e. not what would be anticipated if treatment with spironolactone was rolled out. It was conducted in secondary care. To test the sensitivity of our results to our assumptions made about expected usual care, this secondary care pathway was assumed for a sensitivity analysis. Blood tests were carried out for all participants at baseline only in the clinical trial and those with blood test results suggesting renal problems, relevant comorbidities or on treatments with increased risk were excluded from the trial, explaining why ongoing blood tests were not used. Existing guidelines and expert opinion recommends that in clinical

	practice those women aged >45, with comorbidities or on treatment with increased risk should be monitored. Wording clarified in footnotes of figure 1 to clarify slightly: “†Existing evidence and expert opinion recommends ongoing blood monitoring for women aged >45 years, or those with relevant comorbidities or on treatments with increased risk. As the latter two were not included in the trial, it is not possible to estimate the proportion of such patients that might receive spironolactone and need blood test monitoring. 6/201 (3%) patients in the spironolactone arm of the trial were aged >45 years.”
Reviewer: 2	
Comments to the Author:	
The authors report a cost-effectiveness analysis of the SAFA trial, in which 410 women were randomized to receive spironolactone vs. placebo for 24-weeks. The authors collected detailed cost and quality of life data to inform the CEA. This study has several strengths including its treatment of uncertainty and inclusion of extensive sensitivity analysis to address missing data and lack of standard of care comparator. That said, I have some suggestions to improve the analysis. In particular, I would recommend the authors amend their conclusions to better reflect the uncertainty underlying their cost-effectiveness results (see last comment below).	We thank the reviewer for their useful comments on the paper.
Abstract	
 In reporting results, the authors mention adjusted and unadjusted results, but it is unclear how these results were adjusted in the Abstract methods section. 	We had initially excluded this detail in the abstract methods due to the limited word count. We have now added detail on the variables adjusted for in analyses (and the points below) but it takes our abstract over the word limit. We have been unable to find sufficient words to cut in order to meet the word limit. If the word limit is strict we think this information is the least worst detail to exclude from the abstract as it is clearly stated in the methods of the main part of the paper. See page 3 of the main paper with tracked changes: “Adjusted analysis included randomisation stratification variables (centre, baseline severity [IGA <3 versus ≥3]), and baseline variables (including Acne-QoL symptom subscale score, resource use costs, EQ-5D score and use of topical treatments (Y/N)).”

 Please report what the NICE threshold value is. 	This has been added to the abstract (see page 3 of the main paper with tracked changes).
 NICE acronym is not defined (may be helpful for BMJ Open global audience). 	This has been added to the abstract (see page 3 of the main paper with tracked changes).
 Please do not use the term “cost-ineffective,” rather the intervention was not cost-effective under the NICE willingness-to-pay. 	This has been amended.
Methods	
 Please include the surveys used to collect patient costs as an appendix. 	We have added the questionnaire used at 6 weeks to supplementary materials as an example (see supplementary materials Appendix S2). The resource use questions asked at different time points followed a similar format. This questionnaire also included questions used in the broader study, these are not reproduced here due to length.
 Is assuming no intervention costs for the no treatment group a fair comparison? These people would likely present for care and receive other treatment. I would assume that a proportion would likely be seen by a GP or dermatologist for severe acne even if they opt to not use treatment? This could be a sensitivity analysis. 	The control group did report health resource use related to their acne and this was costed and included in the economic evaluation. Only intervention costs (i.e. placebo drug, visits to issue the drugs) were not included. We have tried to make this clearer in the paper.
 “Several sensitivity analyses were agreed before...” Should include agreed upon. 	Added.
 In the SA comparing spiro to systemic antibiotics, why assume that the systemic antibiotics had QALY effects similar to placebo in the SAFA trial? I would recommend using literature estimates on the effects of systemic antibiotics on acne clearance, or even better HRQoL utility if available. This can then better be compared to spiro, even if the underlying population is different. 	As described above, we would have liked to have used estimates from the literature but could not find appropriate evidence in the literature to use for this or similar populations. Furthermore, from the available evidence using clinical outcomes, there are not clear improvements when comparing oral antibiotics plus topical treatments with topical treatments alone (see details above). We acknowledged in the paper that it is a strong assumption to assume incremental QALYs remains the same as against placebo plus routine topical treatments and that is why we undertook threshold analysis. This shows as reviewer 1 points out that there is a lot of uncertainty around the results of this analysis and is an area that will be strengthened by an ongoing trial with economic evaluation directly comparing spironolactone to oral antibiotics referenced in our paper.
 Is it appropriate to assume the differential missingness was happening at random? One could imagine that those in the placebo group were dissatisfied with their acne treatment 	We have assumed MAR (missing at random), which is defined as the probability that data are missing is independent of unobserved values, given the observed data. We are not assuming MCAR (missing completely at random) – the probability that data are missing is

and therefore dropped out of the study/did not complete study forms.	independent of both observed and unobserved values. We know from looking at the data that missingness is not completely at random because the proportion of missing data is different between the two groups. From the observed data it is not possible to distinguish MAR and MNAR (missing not at random). Multiple imputation is a reasonable model for handling both MAR and MNAR and conducting further sensitivity analyses explores the uncertainty in our data.
 Week 12 patients could take hormonal treatment, oral antibiotics, or isotretinoin 	It is not clear what the reviewer is asking us to change as this point is already in the paper.
 Have the authors considered completing a CHEERS checklist? 	Yes, this was submitted with the paper as supplementary material for the editor.
Results	
 Please clarify the time horizon for the ~0.4 estimated QALYs – assuming it is 6 months. 	This is stated in the methods section: “utility values were used to estimate QALYs over 24 weeks” and has been added to the outcomes section and Table 2
 The authors should note that the 95% CI for the QALY difference between groups included 0. This means that the spiro may impart no difference, or even worse HRQoL utility compared to placebo. I understand this uncertainty is partially accounted for in the PSA, but the authors should highlight this fact in the Abstract, Results, and Discussion. 	It has been argued that decisions should not be based on statistical inference, that is whether mean differences are statistically significant or not. This study was not powered to detect a significant difference in QALYs and therefore the best information we have to make a decision is the mean estimate (Claxton K. The irrelevance of inference: a decision-making approach to the stochastic evaluation of health care technologies. J Health Econ 1999; 18: 341–64). However, the 95% CIs do reflect a high degree of uncertainty and we have now stated this more clearly in the results and discussion section.
 Please correct the reference error on page 8. 	We can only see the following reference on the page mentioned by the reviewer and cannot spot any error with this: “Xu J, Mavranetzouli I, Kuznetsov L, et al. Management of acne vulgaris: summary of NICE guidance. BMJ 2021; 374: n1800”
 It is important that the authors compared spironolactone treatment to “standard of care” oral antibiotics as these would likely be the alternative clinical strategies. The authors should clarify the QALY benefit assumed for oral antibiotics. If it was identical to placebo in the SAFA trial, this is not an adequate comparison. As suggested above, a more appropriate comparison might be to obtain QALY effect estimates from the literature or other sources and compare that to what was estimated for spironolactone. 	See response above and to Author 1 comments.

Discussion	
 “where a more realistic comparator (standard of care based upon NICE guidelines) is used, spironolactone is found to be highly cost-effective” Please clarify the QALY assumptions for SOC treatment and adjust this statement. 	See response to above and to Author 1 comments. We have amended this sentence in the manuscript. See page 10 of the main paper with tracked changes: “In sensitivity analyses, where spironolactone plus routine topical treatment is compared to oral antibiotic plus routine topical treatment (standard of care based upon NICE guidelines) based on the assumptions outlined in the methods section, spironolactone is found to be cost-effective.”
 “On balance, when dealing with missing data or considering a more realistic comparator (providing antibiotic treatment), the likely conclusion is that spironolactone is cost-effective.” I am not convinced by the results that spironolactone is likely cost-effective. The trial did not find a significant difference in QALY effects, and the base-case analysis found spironolactone to not be cost-effective at standard NICE WTP. When including uncertainty, spironolactone is cost-effective in only ~30% of simulations at standard NICE WTP. The authors should base their conclusions on the results of their basecase analysis, and better address the underlying uncertainty. There is likely high value of future information clarifying this uncertainty, in particular comparing spironolactone to oral antibiotics, and future research might better elucidate the cost-effectiveness of spiro for acne treatment. 	We have revised the conclusion to reflect this comment (see last paragraph on page 12 of the main paper with tracked changes) and highlight a number of times the ongoing trials in France and the US which will provide research evidence on the direct comparison of spironolactone and oral antibiotics. It has been argued that decisions should not be based on statistical inference, that is whether mean differences are statistically significant or not. However, as the reviewer suggests, this information can help inform the likely need for further research to reduce uncertainties (Claxton K. The irrelevance of inference: a decision-making approach to the stochastic evaluation of health care technologies. J Health Econ. 1999 Jun;18(3):341-64. doi: 10.1016/s0167-6296(98)00039-3. PMID: 10537899.).

VERSION 2 – REVIEW

REVIEWER	Mavranezouli, Ifigeneia Duke University
REVIEW RETURNED	23-May-2023

GENERAL COMMENTS	The authors have appropriate addressed my concerns.
---

REVIEWER	Borre , Ethan Duke University
REVIEW RETURNED	23-May-2023

GENERAL COMMENTS	The authors have appropriate addressed my concerns.
---

VERSION 2 – AUTHOR RESPONSE

- Reviewer 2 states that we have appropriately addressed their concerns, and many of their concerns overlapped with those of reviewer 1.
- There are some new requested amendments given by reviewer 1, which we feel is inappropriate for a round 2 review.
- We wrote a Health Economic Analysis Plan, approved by the trial management group, ahead of a data analysis and feel that it is important to present the analyses we stated we would undertake and resist undertaking post-hoc analyses.
- The latter is particularly tempting because the results of this analysis are unusually (and interestingly) uncertain, with sensitivity analyses affecting the conclusions in both directions. We have tried to amend the paper to emphasise this point about the uncertainty of the results, over all others. We hope that the amendments we have made improve the paper and that you now consider it suitable for publication in BMJ open in due course.

Many thanks for considering the paper further,

Yours sincerely

Tracey Sach

VERSION 3 – REVIEW

REVIEWER	Mavranezouli, Ifigeneia National Collaborating Centre for Mental Health, Research Department of Clinical, Educational and Health Psychology, University College London
REVIEW RETURNED	27-Aug-2023

GENERAL COMMENTS	I would like to thank the authors for further addressing my concerns. I do understand the limitations of the trial design and the need to adhere to the protocol and I think the manuscript (especially the abstract) and the conclusions are more balanced now and fully reflect the results of the analysis and the underlying uncertainty. My only (minor) suggestion would be to amend the following sentence in the results (under Sensitivity analyses): “With regards to the oral antibiotic control analysis (SA3) [...] at a £30,000 threshold”. I found this rather confusing as the wording (QALY benefit would have to “decrease”, “switch” the ICER) implies that oral spironolactone has been compared with oral antibiotics in the trial. I think the wording has been taken from the previous version of the manuscript, where an ICER of £9,169/QALY was reported for this (hypothetical) comparison, but it is less relevant in the current, further revised version, which only reports the results of the threshold analysis around this comparison. I might say: “[...] the incremental QALY benefit for spironolactone compared with oral antibiotics would have to be 0.00057 (0.000384, MI adjusted) or less, over 24 weeks, for spironolactone
--

	to be less cost-effective than oral antibiotics at a £30,000 threshold”. Alternatively: “the incremental QALY benefit for spironolactone compared with oral antibiotics would have to be 0.00057 (0.000384, MI adjusted) or more, over 24 weeks, for spironolactone to be more cost-effective than oral antibiotics at a £30,000 threshold”. But it’s still fine if the authors would prefer to retain the current wording.
--	--

VERSION 3 – AUTHOR RESPONSE

Response to reviewers:

Reviewer 1	
Comments to the author:	
I would like to thank the authors for further addressing my concerns. I do understand the limitations of the trial design and the need to adhere to the protocol and I think the manuscript (especially the abstract) and the conclusions are more balanced now and fully reflect the results of the analysis and the underlying uncertainty. My only (minor) suggestion would be to amend the following sentence in the results (under Sensitivity analyses): “With regards to the oral antibiotic control analysis (SA3) [...] at a £30,000 threshold”. I found this rather confusing as the wording (QALY benefit would have to “decrease”, “switch” the ICER) implies that oral spironolactone has been compared with oral antibiotics in the trial. I think the wording has been taken from the previous version of the manuscript, where an ICER of £9,169/QALY was reported for this (hypothetical) comparison, but it is less relevant in the current, further revised version, which only reports the results of the threshold analysis around this comparison. I might say: “[...] the incremental QALY benefit for spironolactone compared with oral antibiotics would have to be 0.00057 (0.000384, MI adjusted) or less, over 24 weeks, for spironolactone to be less cost-effective than oral antibiotics at a £30,000 threshold”. Alternatively: “the incremental QALY benefit for spironolactone compared with oral antibiotics would have to be 0.00057 (0.000384, MI adjusted) or more, over 24 weeks, for spironolactone to be more cost-effective than	Thank you for the additional comment, we have amended the sentence as requested. It now reads: “With regards to the oral antibiotic control analysis (SA3), the planned threshold analysis using the complete case, adjusted data found that the incremental QALY benefit for spironolactone compared with oral antibiotics would have to be 0.00057 (0.000384, MI adjusted) or less, over 24 weeks, for spironolactone to be less cost-effective than oral antibiotics at a £30,000 threshold.”

oral antibiotics at a £30,000 threshold”. But it’s still fine if the authors would prefer to retain the current wording.	
---	--